


# Using a Hybrid Optimal Interpolation-Ensemble Kalman Filter for the Canadian Precipitation Analysis

Dikraa Khedhaouiria[1], Stéphane Bélair[2], Vincent Fortin[2], Guy Roy[1], and Franck Lespinas[1]

[1]Meteorological Service of Canada, Environment and Climate Change Canada, Dorval, QC, Canada
[2]Meteorological Research Division, Environment and Climate Change Canada, Dorval, QC, Canada

**Correspondence:** Dikraa Khedhaouiria (dikraa.khedhaouiria@ec.gc.ca)

**Abstract.** Several data assimilation (DA) approaches exist to generate consistent and continuous precipitation fields valuable for hydrometeorological applications and land data assimilation. Usually, DAs are based on either static or dynamic approaches. Static methods rely on deterministic forecasts to estimate background error covariance matrices, while dynamic ones use ensemble forecasts. Associating the two methods is known as hybrid DA and has proven beneficial for different applications

as it combines the advantages of both approaches. The present study intends to explore hybrid DA for the 6-hour Canadian Precipitation Analysis (CaPA). Based on optimal interpolations (OI), CaPA blends forecasts and observations from surface stations and ground-based radar datasets to provide precipitation fields over the North American domain. The application of hybrid DA to CaPA consisted of finding the optimal linear combination between i) an OI based on the Regional Deterministic Prediction System (RDPS) and ii) an Ensemble Kalman Filter (EnKF) based on the 20-member Regional Ensemble Prediction

System (REPS). The results confirmed the known effectiveness of the hybrid approach when low-density observation networks are assimilated. Indeed, the experiments conducted for the summer without radar datasets and for the winter (characterized by very few observations in CaPA) showed that attributing a relatively high weight of the EnKF (50 and 70% for summer and winter, respectively) gave better analysis skills and a reduction of false alarms than the OI method. A deterioration of the moderate to high-intensity precipitation bias was, however, observed during summer. Reducing to 30% the weight attributed

to the EnKF permitted alleviating the bias deterioration while improving skill compared to the OI-based CaPA.

## 1  Introduction

Errors inherent in the observed data, the parameterization of subgrid processes, initial conditions, model physics, to name a few, combined with the chaotic nature of the atmosphere (Lorenz, 1963), lead to uncertain NWP forecasts, especially for precipitation fields (Ebert, 2001). Today, most meteorological centers have developed their operational ensemble prediction

systems (EPS) to consider these uncertainties (Buizza, 2019). EPSs provide, for the same time and location, a set of forecasts obtained by introducing perturbations at different modeling stages (e.g., initial and boundary conditions, Buizza, 2019).

EPSs deliver probabilistic meteorological information that is also valuable for data-assimilation (DA) systems, which will be the focus of this paper for the particular case of the Canadian Precipitation Analysis (CaPA, Mahfouf et al., 2007; Fortin et al., 2015). The CaPA system produces gridded precipitation fields based on numerical weather prediction (NWP) forecasts



adjusted with observed precipitation (ground stations and radars) using optimal interpolation (OI) assimilation methods (Fortin et al., 2015; Lespinas et al., 2015; Fortin et al., 2018). Using EPSs in CaPA would allow the estimate of the 'errors of the day' from short-term ensemble forecasts through the computation of flow-dependent background errors covariances (Kalnay et al., 2007). As opposed to a static assimilation approach, weights attributed to the background and the observations would be potentially better distributed (Wang et al., 2007). However, DA systems using EPSs such as Ensemble Kalman Filter (EnKF,

Evensen, 2003) often require large ensembles to avoid sampling errors and the associated rank problem of covariance matrices (Houtekamer and Zhang, 2016). In contrast, DA systems using deterministic forecasts such as standard variational approaches are more computationally efficient (Wang et al., 2008a; Houtekamer and Mitchell, 1998). Nevertheless, these DAs make hypotheses on the homogeneity, the isotropy, and the invariance of error covariance matrices. To combine desirable aspects of both DA scheme families (Houtekamer and Zhang, 2016), linear combinations of the static and the dynamic error covariance

matrices, known as hybrid approaches, have been developed (Hamill and Snyder, 2000; Wang et al., 2008a; Kleist and Ide, 2015).

Hybrid assimilation schemes have been thoroughly and successfully tested for many applications. Hamill and Snyder (2000) combined 3DVar and EnKF DA schemes for atmospheric assimilation for the United States of America (USA) domain. The authors showed improvements over 3DVar alone, especially in data-poor networks and, to a lesser extent, in denser networks.

For the same domain, Wang et al. (2008a) also showed that analyses generated with the same hybrid approach (3DVar+EnKF) produced 12-hour forecasts that were more accurate than 3DVar and confirmed the efficiency of the method in regions with low observation density. Other studies worked on different hybridization variants, Wang et al. (2008b) combined the Kalman Ensemble Transform Filter (ETKF) and 3DVar for the DA system over the USA using the Weather Research and Forecasting Model (WRF). The authors drew the same conclusions regarding the density of observations. In ocean forecasting, Counillon

et al. (2009) combined optimal interpolation (OI) and an EnKF method. They demonstrated a reduction in forecast errors using hybrid covariances for small dynamic ensembles (10 members) relatively to the OI or EnKF.

The successful combination of two different DA schemes for producing better analysis and forecasts has led various meteorological centers to use this approach for operational products such as Houtekamer et al. (2019) for the Canadian Centre for Meteorological and Environmental Prediction (CCMEP) of Environment and Climate Change Canada (ECCC); Bonavita et al.

(2015) for the European Centre for Medium-Range Weather Forecasts (ECMWF) and Penny et al. (2015) for ocean forecasting at the National Centers for Environmental Prediction (NCEP). Several other studies have shown the relevance of hybrid approaches for DA, but a full review of their uses is beyond this research scope.

At CCMEP, the availability of the Regional Ensemble Precipitation System (REPS) encompassing the North American domain allows the investigation of hybrid approaches for analyses at a higher resolution than those currently existing (see

Houtekamer et al., 2019, for the global domain with its horizontal grid spacing of about 39km). The 20-member REPS product is available at an approximately 10-km grid spacing and has been operational since summer 2019. The purpose of this study is to explore the potential improvements brought by hybrid approaches to the operational 6-hour CaPA, which is solely based on an OI where the background field is the Regional Deterministic Prediction System (RDPS, Caron et al., 2015). Various factors motivated the exploration of the well-established and documented hybrid DA methods. First of all, the REPS is fully





operational and has the same domain and resolution as CaPA and, therefore, can be used as a background field without any interpolation. Second, the hybrid approaches may positively impact the several data-poor areas (e.g., northern Canada) of the CaPA domain where the OI has its limitations. Thirdly, to our knowledge, no other studies have explored this approach for precipitation analyses. Finally, the OI's specificity in CaPA makes it interesting to explore in the context of hybrid DA approaches. Indeed, conclusions from several studies on hybrid DA methods were generally obtained when combining static

and dynamic DA approaches (Wang et al., 2007; Counillon et al., 2009). However, the OI in CaPA is not strictly speaking static as observations, and forecasting errors are updated for each analysis time using variographic analysis (for more details see Fortin et al., 2015) as opposed to static errors estimated with the climatology.

The rest of the article is organized as follows. Section 2 introduces the hybrid analysis scheme, while Section 3 describes the method to select the optimal weighting for the hybrid approach. Section 4 and 5 present respectively the datasets and

experimental design followed by the verification strategy in Section 6. The results of the experiments are made available in Section 7 and the conclusion and the discussions are given in Section 8.

## 2   Hybrid assimilation approach for the precipitation analysis

Similarly to what was proposed by Hamill and Snyder (2000), the the background field error covariance matrix using the hybrid approach ($\widetilde{P}^b$) is a weighted sum of the OI-based ($P^b_{OI}$) and dynamically-based ($P^b_d$) matrices such as:

$$\widetilde{P}^b = (1-\beta)P^b_{OI} + \beta P^b_d, \qquad (1)$$

where the parameter $\beta$ is comprised between 0 and 1, ensuring that the total background error covariances are conserved (Wang et al., 2008a). Thus, when $\beta=0$, the analysis is solely based on the OI, while when $\beta=1$ the background errors are ruled by the REPS. The $\beta$ selection methodology is further detailed in Section 3.

It is worth mentioning that the precipitation from both the observation and background model ($y$) are primarily Box-Cox

transformed such as:

$$x = \begin{cases} \lambda^{-1}\left[y^\lambda - 1\right], & \text{if } \lambda > 0, \\ \log(y), & \text{if } \lambda = 0, \end{cases} \qquad (2)$$

and where in CaPA configuration $\lambda = 1/3$. This pre-processing strategy has been used as it provided more skillful analyses (Fortin et al., 2015; Lespinas et al., 2015). The final analyses are obtained from the back-transformation, followed by corrections of the biases induced by this transformation (Evans, 2013).

An exponential and isotropic model is assumed for the spatial correlation function of the background field errors (Fortin et al., 2015). The $i$-th row and $j$-th column of $P^b_{OI}$ writes as:

$$p^b_{OI}(i,j) = \sigma^2_{OI}\exp(-\delta(i,j)/l_{OI};) \qquad (3)$$

where $\sigma^2_{OI}$, $\delta(i,j)$, and $l_{OI}$ are, respectively, the variance of the background errors, the Euclidean distance between locations $i$ and $j$ and the correlation length. These parameters are estimated using variographic analysis of the innovations, $Z = d - Hx_f$.





The $d$, $x_f$ and $H$ correspond respectively to the measurements, the forecasts and the observation operator, which is here the nearest neighbour interpolation (Fortin et al., 2015).

The degree of spatial dependence of the innovations is described through a theoretical exponential function fitted on the empirical semivariogram (Cressie, 2015) and is defined as:

$$\gamma_{i,j} = \frac{1}{2}\left\langle |z_i - z_j|^2 \right\rangle = \begin{cases} \sigma_o^2 + \sigma_{\text{OI}}^2 (1 - \exp\left(-\delta\left(i,j\right)/l_{\text{OI}}\right), & \text{if } \delta\left(i,j\right) > 0, \\ 0, & \text{otherwise.} \end{cases} \tag{4}$$

where $\sigma_o^2$ corresponds to variance errors of the observation. As opposed to other DA approaches where the error matrix is static, the variographic analysis is computed for each analysis time step and therefore allows for time varying elements in $\text{P}^a{}_{\text{OI}}$.

The covariance matrix $\text{P}^b{}_d$ depicts the flow-dependant errors estimated from the REPS and is defined as:

$$\text{P}^b{}_d = \frac{1}{N-1} \text{A}' \text{A}'^{\text{T}}, \tag{5}$$

where $\text{A}' \in \mathbb{R}^{m \times N}$ denotes the anomalies estimated from the $N$-member ensemble at $m$ grid points. The superscript T corresponds to a matrix transpose. To avoid underestimation of the variance of the background errors due to the limited size of the dynamical ensemble (Houtekamer and Mitchell, 1998), the anomaly computation follows Hamill and Snyder (2000) suggestions and writes as:

$$\text{A}'_{:,j} = \text{A}_{:,j} - \overline{\text{A}}_{[\![1,N]\!]\setminus j}, \tag{6}$$

where $\text{A}'_{:,j}$ is the $j$-th column of $\text{A}'$ and $\overline{\text{A}}_{[\![1,N]\!]\setminus j}$ is the average across the $N$ members without the $j$-th member.

The analysis is performed sequentially for each grid-cell, for which the error matrices are estimated using up to 16 neighbors per observation type to speed up the computation (Fortin et al., 2015). Therefore, when assimilating both surface observations and radar Quantitative Precipitation Estimates (QPEs), $\widetilde{\text{P}}^b$ is a matrix of size up to $32 \times 32$. In cases where no observation within a radius of $500\ \text{km}$ is encountered, the analysis process is accelerated by setting the grid-cell value equal to the background field.

The equations for the analysis estimates are solved as:

$$\begin{aligned} x_a &= x_f + \text{W} \cdot (d - \text{H}x_f), \\ \text{W} &= \widetilde{\text{P}}^b \text{H}^{\text{T}} \cdot \left( \text{H} \widetilde{\text{P}}^b \text{H}^{\text{T}} + \text{R} \right)^{-1}, \end{aligned} \tag{7}$$

where $\text{R}$ and $\text{W}$ are, respectively, the observation error covariance matrix (see details in Fortin et al., 2015) and the weight vector.

The variances of the error made by estimating the precipitation analysis is obtained from the covariance structure and the weights as follows (Fortin et al., 2015):

$$\sigma_a^2 = \sigma_b^2 - \text{W}^{\text{T}} \cdot \widetilde{\text{P}}^b \text{H}^{\text{T}} \tag{8}$$

For the hybrid approach, the variance of background errors is defined as:

$$\sigma_b^2 = (1-\beta)\sigma_{\text{OI}}^2 + \beta \sigma_{b,\text{REPS}}^2, \tag{9}$$





where $\sigma^2_{\text{OI}}$ is estimated with variogram modelling presented in equation 4. In contrast, $\sigma^2_{b,\text{REPS}}$ has the advantage of being
directly estimated from the REPS and defines for a given grid-cell location, $s_o$, such as:

$$\sigma^2_{b,\text{REPS}}(s_o) = \frac{1}{N-1}\left(x_{s_o}^{(\text{REPS})} - \overline{x}_{s_o}\right)\left(x_{s_o}^{(\text{REPS})} - \overline{x}_{s_o}\right)^{\text{T}},\tag{10}$$

where $x_{s_o}^{(\text{REPS})}$ is the N-length vector of REPS precipitation at location $s_o$ and $\overline{x}_{s_o}$ is the ensemble mean at the same location.

In addition to the estimate of the analysis error already provided by the OI, CaPA provides a spatially and temporally varying
index that describes the confidence granted to the precipitation analysis. This index is based on the assumption that the most
trustful data is observation. For a given grid-cell and valid time, the confidence index of the analysis (CFIA) has been defined
as one (1) minus the ratio of the error variances of the analysis, $\sigma^2_a$, to the background, $\sigma^2_b$, such as:

$$\text{CFIA} = 1 - \frac{\sigma^2_a}{\sigma^2_b}\tag{11}$$

The CFIA ranges from 0 to 1. CFIAs close to 1 (0) depict high contributions of the observation (background) into the analysis
estimates. CFIAs close to 0 occurs when no observations are assimilated in the vicinity of a given grid-cell. Therefore, this
index is convenient for users desiring to select only grid-cells influenced by observations. Using the hybrid approach in CaPA
enables to account for flow-dependent errors. It would therefore have impacts on CFIA estimates discussed in Section 7.4.

## 3 Selection of the weighting factor: $\beta$

Analysis experiments using $\beta \in [0,1]$ with a 0.1 step are conducted for each valid time to identify the $\beta$ value that provides
the most accurate precipitation analysis. At least two different options are possible to select the most suitable $\beta$. First, the
analyses are conducted for a given season, and the $\beta$ that minimizes an objective function is selected and stored for future
uses. The second option consists of choosing the most suitable $\beta$ for each valid time, implying a dynamical selection of the $\beta$.
While the first option is computationally inexpensive as $\beta$s are estimated once for each season, it suggest that this parameter
remains constant over years. The second option is more flexible but needs independent data from those which relied upon for
the precipitation verification step, which is required to evaluate the CaPA system (Section 6).

To understand the choice that has been made for either option, the CaPA configuration must be presented first. CaPA gen-
erates two types of precipitation analysis. The first is conducted only at surface station locations in a leave-one-out (LOO)
framework (hereafter, CaPA-LOO). This computationally inexpensive analysis is both a quality control step to reject suspi-
cious precipitation observations (see Section 2.b in Lespinas et al., 2015, and Section 4.2 below) and a valuable dataset for
verification purposes in a cross-validation manner. The second type of analysis is performed for each grid-cell of the domain.

A dynamical selection of $\beta$ for each analysis valid time would have been the preferred option as it would have not required
manual updates during, for example, major operational upgrades of the different systems (CaPA, REPS, or RDPS). For this
purpose, preliminary tests using random samplings of the CaPA-LOO datasets have been done. For each valid time, a training
sample was employed for the $\beta$ selection, while the testing sample was kept for objective verification. The training and the
testing sample sizes had to be large enough to, respectively, capture the optimal $\beta$ value over the domain and to realize





consistent precipitation verification. Different sample sizes were tested with up to 40% of the initial CaPA-LOO dataset to estimate the optimal $\beta$. However, the results were too noisy due to sampling effects, especially during winter where the station density is relatively low. For this reason, the first option to select $\beta$ value was preferred. The normalized root mean square error (NRMSE) averaged over a selected period and based on the CaPA-LOO datasets was chosen to identify optimal $\beta$. The normalization was realized to lessen the influence of the total precipitation amount during a given analysis time step (Bachmann et al., 2019). For more reliability, the NRMSE is computed only at SYNOP and manual SYNOP stations during the summer and the winter, respectively, and is defined as follow:

$$\text{NRMSE} = \frac{\sqrt{\sum_{k=1}^{M} \left(a_k^{\text{LOO}} - o_k\right)^2}}{\sqrt{\sum_{k=1}^{M} \left(a_k^{\text{LOO}} + o_k\right)^2}}, \tag{12}$$

where $M$, $a^{\text{LOO}}$ and $o$ correspond, respectively, to the number of cases during a given valid time, the LOO analysis, and the observations. The NRMSEs of zero (0) illustrate perfect analyses.

## 4 Datasets

### 4.1 Model description

The operational 20-member REPS (version 3.0.0, ECCC, 2019) and the Regional Deterministic Prediction System (RDPS, Caron et al., 2015) use the same configuration. The domain covers the North American continent (Figure 1) with a horizontal grid spacing of 0.09° ($\sim$ 10-km) and 84 vertical levels. The RDPS/REPS generate 72-h forecasts four times per day at 00, 06, 12, and 18 UTC. Uncertainty in the REPS is represented by perturbed initial and lateral boundary conditions (ICs and LBCs) and stochastically perturbed physics tendencies (SPPT, Charron et al., 2010) but with the same physical parameterization for all members. The atmospheric ICs derives from a 20-member interpolated Global Ensemble Prediction System (GEPS, Charron et al., 2010; Houtekamer et al., 2014) analysis perturbations centered around the RDPS initial analysis. Every hour, the LBCs are also provided by the GEPS.

### 4.2 Observations

The 6-hour analysis assimilates precipitation from surface stations from Canadian and the contiguous United States (US) networks, some of which are operational only during the warm season. Thirty-three and 31 C-band radar QPEs, covering the US and Canada (mostly located along the US border), are also assimilated. New Canadian dual-polarization Doppler radars have been progressively added to the observation database. They contribute to retrieving information on a broader range of meteorological events than standard C-band radars (Kollias et al., 2020), and would ultimately enable better analysis.

Observed precipitation and radar QPEs undergo an extensive quality control (QC) process to remove untrustworthy data. The QC is automatically done for each assimilation valid time, leading to a time-varying number of assimilated observations as described in Lespinas et al. (2015) (for surface observations) and in Fortin et al. (2015) (for radar QPEs). A first temporal



QC is performed to identify persistent problems that occur over a given period, such as a station that reports no or too much
precipitation over a long time. Rejecting or keeping a surface station is based on the statistical distribution of the differences
between the observation and the analysis obtained over the most recent cases (see details in Section 2.b. in Lespinas et al., 2015).
A second quality control, hereafter referred to as spatial QC, is carried out to identify surface stations that have very different
precipitation than those in the immediate vicinity. For this purpose, an analysis is estimated at a site $s_k$ using neighboring
stations in a LOO approach. The observation is rejected or, is said invalid, if:

$$\left| x_{s_k}^{(OBS)} - x_{s_k}^{(CaPA)} \right| < \text{tol} \cdot \sqrt{(\sigma_o^2 + \sigma_a^2)}, \tag{13}$$

where tol is a tolerance factor set equal to 4 for the operational CaPA, and obs and CaPA superscripts refer to, respectively,
the observation and the analysis in the transformed space. The size of the neighborhood depends on the background field
correlation length which varies for each analysis and therefore adapts with seasons (Lespinas et al., 2015). This approach helps
avoid rejecting very localized summer precipitation events. An additional QC is also applied during the cold season, with the
rejection of radar QPEs and surface observations during windy condition (Rasmussen et al., 2012). Figure 1 illustrates the
stations and radars that passed the QC for summer.

In CaPA, hybrid DA approaches can impact the quality control of observations. Indeed, changing the past analysis values
(used in the temporal QC) and the standard deviation of the analysis error (used in the spatial QC) can induce differences in
the number of assimilated stations. The results showed slight changes in the number of observations assimilated when using
hybrid approaches and are therefore not detailed in this study.

### 4.3 Stage IV precipitation

In addition to the LOO verification at station locations, the 6-hour analyses were also compared to 6-h Stage IV (ST4) analysis
from the National Centers for Environmental Prediction (NCEP, Lin and Mitchell, 2005). The objective is to allow verification
against seamless precipitation fields. ST4 is a mosaic of regional multi-sensor (gauges and WSR-88D radars) analysis that is
designed differently from CaPA and ensures a certain degree of independence during the verification. The ST4 domain covers
the contiguous United States (CONUS), but for robustness reasons, only the CONUS east of 105W was used for verification
purposes (e.g., Nelson et al., 2016; Schwartz, 2019). The ST4 native horizontal spacing of $\sim 4.7$ km was interpolated to the
coarser RDPS grid using circular filtering (see Fig 3.a in Jacques et al., 2018) to allow verifications on a common grid. ST4
will not provide a picture of performance over the entire RDPS domain, as Canada and the southern part of the RDPS are not
covered. However, ST4 is a valuable dataset that could help compare the results obtained with the different $\beta$ values in the
hybrid approach.





## 5   Experimental set-up

Two sets of two-month experiments were conducted: i) one over July to August 2019 (hereafter summer), and ii) one over January to February 2020 (hereafter winter). For each 6-hour valid time, 11 sets of analyses were produced with the different
$\beta$ values presented in Section 3.

Summer and winter experiments are expected to differ due to both the background model's distinct seasonal performances in representing precipitation and the significantly smaller assimilated datasets in winter. The usefulness of a hybrid configuration has been demonstrated when low-density observation networks are assimilated (Hamill and Snyder, 2000; Wang et al., 2008a, b). Thus, to verify this point without being strongly influenced by seasonal effects, an additional experiment was con-
ducted without the assimilation of radar QPEs. The experiment without radar was only conducted during the summer as no impact is expected during the cold season, during which radar QPEs are not assimilated.

## 6   Verification strategy

### 6.1   Comparisons against observation from surface stations

Based on the 2x2 contingency table for binary events (Table 1), four different metrics, commonly used for precipitation objec-
tive evaluation, are computed for the different configurations of the LOO analysis. First, the frequency bias index (FBI) which compares the frequency of events in the analyses to those in the observations. Second, the equitable threat score (ETS) which assess the agreement between the analysis and the observations. Following Table 1 annotations, FBI and ETS express as:

$$\text{FBI} = (a+b).\,(a+c)^{-1},$$
$$\text{ETS} = \frac{a - h_r}{a + b + c - h_r},$$
(14)

where $h_r = (a+b)\,(a+c)$ illustrates the hits expected by chance and make the ETS less sensitive to the climatological fre-
quency of precipitation events. The probability of detection (POD) and the false alarm ratio (FAR) are the other two metrics and define as:

$$\text{POD} = a \cdot (a+c)^{-1},$$
$$\text{FAR} = b \cdot (a+b)^{-1},$$
(15)

FBI-1 – which is equivalent to the normalized difference between false alarms and missed events – is preferred in the following as positive (negative) values imply positive (negative) bias (Lespinas et al., 2015). FBIs-1 and FARs of zero (0) are optimal,
while ideal PODs and ETSs are one (1). For the four metrics, binary events define as 6-hour accumulations meeting or exceeding selected thresholds, here 0.2, 1.0, 5.0 and 10.0 mm. Higher thresholds are not shown because the samples were too small, leading to overly noisy scores.





Statistical differences between the scores using $\beta = 0.0$ (hereafter the reference experiment) and $\beta > 0.0$ were assessed using a stationary block bootstrapping with a 95% confidence level (Brown et al., 2012). The bootstrapping implementation
specific to CaPA is detailed in Lespinas et al. (2015).

### 6.2  Comparisons against Stage IV

Two different types of comparison against ST4 were conducted. First, aggregated areal coverages of 6-h accumulated precipitation meeting or exceeding selected accumulation thresholds were assessed to evaluate how precipitations are distributed in ST4 and CaPA.

Second, the Fraction Skill Score (FSS) was computed to assess the analysis skill in spatially placing precipitation events (Roberts and Lean, 2008; Schwartz et al., 2009). A selected threshold is first applied to each grid-cell of CaPA and ST4 to define the occurrences of precipitation events for a given valid time and for each grid-cell. Then, the fractions of grid-cells above the threshold (probabilities) in a pre-selected neighborhood (for example, a square of 30 km) are calculated for CaPA ($f_a$) and ST4 ($f_o$) respectively. The FSS compares the differences of fractions to the largest possible fraction difference and
expresses as:

$$\text{FSS} = 1 - \frac{\frac{1}{N_y} \sum_{i=1}^{N_y} \left( f_a(i) - f_{o(i)} \right)^2}{\frac{1}{N_y} \left[ \sum_{i=1}^{n} f_{a(i)}^2 + \sum_{i=1}^{n} f_{o(i)}^2 \right]} \tag{16}$$

where $N_y$ is the number of grid-cells on the verification domain. The FSS is averaged over the period and was calculated for squares of 20, 30 and 50 km. Values close to 1 of the FSS are optimal, while 0 indicates no skills. By construction, as the size of the neighbourhood increases, FSSs also increase because overlaps of precipitation events in the observed and analysis
datasets are more likely.

## 7  Results and discussion

### 7.1  Selection of the optimal $\beta$ value

Figure 2 shows the NRMSE estimated at the SYNOP stations for the different $\beta$ values and in the LOO framework. The presence of a minimum at $\beta > 0.0$ for the three experiments, in summer with and without radar QPEs and in winter, demonstrates
the usefulness of the hybrid approach.

Comparing the two summer experiments (Fig. 2.a and b), it can be first noticed that, as expected, the addition of radar QPEs generally reduces the NRMSE and thus improves the performance of the analysis. It also decreases the added value of the hybrid approach. NRMSE values for $\beta$ ranging from 0.0 to 0.4 were indeed very similar for the summer experiment assimilating radar QPEs (Fig 2.b), with a minimum obtained at 0.4. On the other hand, the experiment without radar QPEs
showed larger variability in NRMSEs for the different $\beta$s with a minimum obtained at 0.5 (Fig 2.a). These results suggest that when the density of assimilated observations is lower, the hybrid approach brings more added value and is thus consistent with





the literature (Hamill and Snyder, 2000; Wang et al., 2008a). During the summer, the use of $\beta > 0.6$ (0.9) with (without) radar assimilation deteriorated the NRMSE values compared to the reference analysis ($\beta = 0.0$).

Interestingly, the winter experiments illustrated a different pattern. According to the NRMSEs (Fig 2.c), the analysis improved when the $\beta$ increased and reached a minimum at $\beta = 0.7$. The analysis deteriorated for even higher $\beta$s but was not worse than the reference experiment, meaning that in that case, the dynamic approach was more suited than the static one. The low density of assimilated observations during the winter solid precipitation compared to liquid phase precipitation season may partly explain this result and is consistent with the experiment with and without radars. Another point that could explain this performance is the higher winter forecast skill of both the RDPS and the REPS that allow more accurate background fields and a better specification of the $\widetilde{P}^b$ matrix.

Both the seasons and the quantity of assimilated observations seem to have an influence on the optimal value of $\beta$. Summer 2019, with and without radar QPEs, and winter 2020 have optimal $\beta$ that is equal to 0.5, 0.4, and 0.7, respectively. Further evaluation metrics to verify that these optimal values contribute to improve different aspects of the 6-hour precipitation distribution are presented in the following section.

## 7.2 Contingency table verification

Metrics based on the contingency table, i.e. FBI-1, ETS, POD and FAR (Section 6.1) were calculated for the three experiments and for all $\beta$ values. Attention was paid to the experiments with the optimal $\beta$ according to the NRMSE presented in the previous section (see Fig 2). However, experiments with different $\beta$ values were also investigated because they seemed to provide a good compromise, improving some metrics while deteriorating very little the other. Results with $\beta$ values that degraded too much the reference analysis are not presented.

Figure 3.a illustrates the metrics for the summer without the radar QPEs for the optimal $\beta = 0.5$ compared to $\beta = 0.0$, where filled markers indicate no significant differences at the 95% confidence level between the two experiments for a given threshold. The 6-h precipitation analysis displayed a significant increase of skill (at the 95% confidence level) as shown by the ETS and a decrease of the false alarms (FAR) for all the selected thresholds. The POD was slightly deteriorated, especially when looking at the small precipitation events. As illustrated by the FBI-1, the selection of $\beta = 0.5$ led to generally lower precipitation amounts than with the use of $\beta = 0.0$. The impact was positive for small precipitation events (thresholds of 0.2 and 1.0 mm), but it tended to smooth out higher intensity events. Looking at other $\beta$ values, $\beta = 0.3$ (Fig 3.b) seemed to be a good compromise between skill and bias of the precipitation analysis. Indeed, the ETS and FAR remained improved compared to the reference experiment, and the deterioration of the POD was less important than with $\beta = 0.5$, while the degradation of FBI-1 for heavy precipitation was acceptable.

The results for the analysis during the same season but with the assimilation of the radar QPEs is showed in Figures 3.c and d. Similarly to what was observed for the NRMSE values, all metrics were generally improved when assimilating the radar QPEs but the differences between the reference and the optimal $\beta = 0.4$ were less pronounced. The analysis with $\beta = 0.4$ showed significantly reduced FARs for all thresholds. However, the improvement in ETS when compared to the $\beta = 0.0$ was significant only for small thresholds (0.2 and 1.0 mm). For higher thresholds, the skill was slightly improved but was not




significant at the 95% confidence level. Similarly to the experiment without radar QPEs (Fig 3a. and b.), the POD slightly was slightly deteriorated, and the FBI-1 reduced for small precipitation events but increased for events of medium to high intensity. The use of a $\beta = 0.3$ seemed again to be a good compromise regarding the bias by reducing the analysis smoothness when compared to $\beta = 0.4$, while preventing the POD deterioration.

Finally, Figure 4.a illustrates the same metrics during the winter and compares the $\beta = 0.7$ to the reference experiment. The ETS, was significantly improved, and the false alarms at the 95% confidence level were reduced. Fewer precipitation events were generated for all selected thresholds with the analysis using $\beta = 0.7$ than with $\beta = 0.0$. Again, this improves the performance for 6-hour precipitation greater than 0.2 and 1.0 mm, but not for accumulations greater than 2.0 mm. However, the degradation of FBI-1 for high-intensity precipitation was much less pronounced than in summer, especially for such a high

$\beta$ value. The probability of detecting events (POD) greater than 0.2 mm was significantly reduced, but was increased for heavy events precipitation ($> 10$ mm).

    For the sake of comparison with the summer season, contingency table metrics using $\beta = 0.3$ are also displayed in Fig. 4.b. It appeared that this value also provides improved metrics even though it was not identified as optimal. Indeed, FBI-1 was still, but to a lesser extent, reduced for precipitation above 0.2 mm compared to the reference experiment. It is interesting to note

that for other thresholds, the degradation of FBI-1 was less pronounced and was not significant in most cases. Similarly, the POD degradation for the 0.2 mm threshold was less pronounced. The ETS and FAR were still significantly improved for all thresholds relative to the reference experiment, but to a lesser extent than with $\beta = 0.7$.

    In light of these results, it appears that the use of the optimal $\beta$ value identified through the use of NRMSE did indeed show an improvement in skills and a reduction in false alarm rates for both summer and winter experiments. However, the

frequency of moderate to high intensity 6-hour precipitation was reduced, especially in summer. The latter can be detrimental to very localized high-intensity precipitation events. The examination of the same scores for other values of $\beta$ highlighted that the use of $\beta = 0.3$ offered a good compromise for skill gain while not overly damaging the frequency bias for high-intensity precipitation events for both seasons. For these reasons, $\beta = 0.3$ was maintained as optimal in the sense of a compromise and is used for the additional checks presented below.

**7.3    Verification against ST4**

The impact of radar QPE assimilation in the hybrid approach has been demonstrated in the previous sections. Therefore, to simplify the discussion, the comparison of CaPA analyses with ST4 is only performed for the experiments that assimilated the ground-based radars (i.e., excluding the summer without radar QPE). This point is also justified by the fact that the operational CaPA is integrated with the radar QPEs and that the final objective is to compare with this system. Thus, in the following, the

summer experiments refer to the one that assimilates the radar QPEs.

    Figure 5 shows the accumulated precipitation for ST4 and CaPA using $\beta = 0.3$ and the differences between these two datasets. More spatial patterns and higher summer precipitation amounts were observed in ST4 than in the CaPA using $\beta = 0.3$ (Fig. 5.a). The higher native resolution of ST4 and the limited ability of the CaPA background model to predict convective precipitation, especially at the 10 km grid spacing, may explain such results.





The winter (Fig 5.b), generally characterized by spatially larger precipitation events, showed a better agreement between CaPA and ST4. Indeed, ST4 and CaPA with $\beta = 0.3$ had similar spatial patterns with both over- and underestimated accumulations depending on the location, but to a lesser extent than in summer. These results can be interpreted both by the greater forecasting capacity of the background model in winter and by the expected lower impact of the horizontal resolution during this season. Accumulated precipitation, using either $\beta = 0.0$ or $\beta = 0.3$, was compared to ST4 at the grid point scale to identify

the best parameter (not shown for conciseness). It was found, that without spatial consistency, some locations (grid cells) were better represented with $\beta = 0.3$ while for other locations $\beta = 0.0$ was more suitable. Therefore, moving from $\beta = 0.0$ to $0.3$ values have a minimal impact on seasonal precipitation accumulations.

    Figures 6.a and 6.b depict the fraction of grid-cells above selected 6-h thresholds averaged for the summer and winter, respectively. During summer, the differences between the configurations $\beta = 0.0$ and $\beta = 0.3$ were very small ($< 2$ %), as

shown by the very close golden and blue curves. Consistent with results on the FBI-1 (Fig. 3), the fraction of precipitation events above $0.2$ mm generated in the analysis using $\beta = 0.0$ or $0.3$ were slightly higher than in ST4 for both season. During summer, the fraction of grid points greater than 1, 2 and 5 mm.6h$^{-1}$ were systematically underestimated by 2 to $3\%$ in CaPA compared to ST4 and these fractions appear to contribute importantly to the underestimation of ST4 precipitation totals (Fig. 5). Winter was similar (Fig 6.b) as fractions of grid-cells for the 6-h precipitation analysis were generally overestimated in the

analysis. However, this overestimation was relatively smaller ($< 1\%$) than summer and became almost negligible for 5 and 10 mm.6h$^{-1}$ thresholds.

    The FSSs for the 11 analyses differing by $\beta$ values and using ST4 as a reference were calculated for the 0.2 and 1.0 mm.6h$^{-1}$ thresholds with neighborhood length scales of 20, 30, and 50 km (Fig. 7). For a given neighborhood length scale and for the 0.2 mm threshold during the summer (Fig. 7.a), the FSS increased when $\beta$ went from 0.0 to 0.5 and then decreased for higher

$\beta$s, thus are consistent with the results obtained with NRMSEs. Similarly, during the winter and for the same threshold, FSSs increased from when $\beta$ went from 0.0 to 0.7 and decreased thereafter (Fig. 7.c). The gain associated with the use of $\beta > 0.0$ was less marked for the 1.0 mm threshold, especially for the summer (Fig. 7.b). These results are consistent with those obtained with scores based on the contingency tables for the summer experiment. For both seasons (Fig. 7.b and 7.d), the use of $\beta = 0.3$ did not, however, deteriorate the FSSs compared to those of the reference experiment and even showed a slight improvement

during the winter.

### 7.4   Confidence index

In the OI-based CaPA, CFIA fields are characterized by circular structures as shown for two winter cases in Figure 8.b. Small dots of high values are centered around the surface stations and large disks represent radar footprints. These particular structures are a direct consequence of the assumptions on the error isotropy.

Using the hybrid approach in CaPA enables to account for flow-dependent errors, allowing CFIA spatial distributions to be linked to meteorological situations and thus lessen the influence of the aforementioned assumptions. However, in this configuration, the variances of the background field errors are also spatially variable, in contrast to the spatially constant value used for operational CaPA (eq. 4 and 9). Therefore, this makes the interpretability of CFIA, in its current definition




(eq. 11), less straightforward. To illustrate this point two different meteorological situations were selected and are illustrated
in Figures 8 (a,d). The 6-h precipitation fields for January 5, 2020 (Fig 8.a) and January 18, 2020 (Fig 8.d) valid at 12UTC,
were both characterized by large synoptic events but at different locations. Increasing the contribution of the REPS in the
analysis computation, $\beta = 0.7$ in Fig 8 (c,f), leads to CFIAs following meteorological spatial distributions. Generally, CFIAs
tended to be higher at places with precipitation than when using $\beta = 0.0$ as shown in the eastearn part of the domain for the
January 18 case (Fig 8.f). Nonetheless, CFIA remained close to zero above the Atlantic ocean for the case of January 5 (Fig
8.c) despite the synoptic event that occurred in that location. This is explained by the current CaPA computation framework,
where gridcells are set equal to the background when no observations are available in the vicinity. As a result, the current CFIA
(eq. 11) revealed two limitations for the hybrid assimilation approach: (i) users interested in gridcells with high CFIAs may
indirectly favor locations with precipitation which in turn can lead to biased interpretation and (ii) low CFIA values may have
two different interpretations. To overcome the first limitation, a field representing the density of assimilated observations for
each gridcell will come along with the CaPA precipitation fields. In addition, to ease CFIA interpretation, the reference, which
is currently $\sigma_b^2$ in eq. 11, will be changed for a temporal climatology of the background field errors. To do this, the denominator
in the equation 11 will be replaced by the time average of $\sigma_b^2$, calculated for each grid cell using an exponential filter that gives
more weight to the most recent cases. Different tests made on CFIA estimates are beyond the scope of this paper and will be
discussed in a different study.

## 8   Conclusions

Based on optimal interpolation, the Canadian Precipitation Analysis, currently operational at CCMEP, is used for different
applications (see Fortin et al., 2018, for further details). The recently available operational 20-member REPS with a $\sim 10\,\mathrm{km}$
grid-spacing naturally led to examine whether it was possible to improve the estimation of background error covariances used
by the current CaPA OI. From this perspective, the selected approach is a so-called hybrid approach (Hamill and Snyder,
2000; Counillon et al., 2009) and it consists of a weighting of two covariance matrices, one obtained by the OI and the other
from an ensemble Kalman filter based on the REPS. The primary motivation is to overcome some simplifying assumptions
about the structure of background errors when using the OI. Indeed, the operational CaPA uses a spatial covariance structure
of background errors assumed isotropic and is modeled as an exponential function. The second motivation is the documented
positive impact of this approach for domains with low observation density, which is the case for some regions of the CaPA
domain (e.g., northern Canada) and during the winter season when few stations are assimilated.

In practice, this consists of choosing an optimal weighting between two covariance matrices of background field errors,
one from the OI and the other from REPS, through a parameter – $\beta$ – which varies between 0 and 1. The experiments were
conducted for 6-hour precipitation accumulations and over two seasons, summer 2019 and winter 2020. To verify the impact of
the amount of assimilated observations, an additional experiment without radar QPEs' assimilation was conducted for summer
2019. This additional experiment was not performed for winter as the assimilated radar QPEs during the solid precipitation
season are anyway negligible in CaPA (Fortin et al., 2015).



Results have shown that first, the analysis with the assimilation of the radar QPEs limits the positive impacts of the increase of the weighting towards the REPS (i.e., high $\beta$s). Indeed, with and without the radar QPEs assimilation, and according to NRMSE estimates, summer had optimal values of 0.4 and 0.5, respectively. Winter precipitation showed higher optimal values of $\beta$, around 0.7, meaning that a higher weighting in favor of the REPS is preferable during that season. This result confirmed the known positive impact of the hybrid method in the configuration of a low station density. The second reason is the higher forecasting skill of the background model (RDPS) and the REPS during the winter season, leading to a better error specification. Interestingly, these optimal values obtained when comparing precipitation from surface stations to the analysis (in a leave-one-out framework) were consistent with those obtained using the Fractions Skill Scores, using a relatively independent observation dataset and on a smaller domain, the Stage IV analysis (Lin and Mitchell, 2005).

The use of the hybrid approach with the optimal $\beta$ values for each experiment increased the analysis skills of both the operational CaPA ($\beta = 0.0$) and a purely dynamic approach ($\beta = 1.0$). This result is especially true for low-intensity precipitation events, for which the false alarm rate has been significantly reduced. In summer, the frequency bias score was reduced for small precipitation events (up to 1 mm) but degraded for higher thresholds. High-intensity precipitation events were indeed smoothed when using the optimal $\beta$ value. This result was also observed for the winter but to a lesser extent. Using a trade-off value of $\beta = 0.3$ for the summer experiments reduced the impact of smoothing high-intensity events while allowing for significant skill improvement. According to the winter 2020 experiment results, a higher $\beta$ would have been more suitable. However, in practice, this means choosing a date in the CaPA configuration for which $\beta$ must be modified. A more extended test period and further experiments would be needed to distinguish whether the parameter changes should be made during the regular North American domain seasons or during precipitation phase changes (approximately in May and October). In the meantime, other tests, using $\beta = 0.3$ during the winter, have provided acceptable results in skill and bias while being conservative.

Different flavors of the hybrid approach have also been tested but have not been developed in this article as they did not contribute positively or significantly to the selected metrics. As suggested in Counillon et al. (2009) study, an inflation factor ranging from 1.0 to 2.0 was applied to alleviate the possible lack of spread in the REPS but the experiments showed no or negative impacts on the verification metrics. Another hybrid configuration was also conducted to mitigate the impact of high-intensity events smoothing during summer precipitation. For that test, the hybrid approach was activated only if at least half of the REPS members had strictly positive precipitation; otherwise, $\beta$ was set to zero. The resulting scores showed higher biases for low-intensity precipitation events than the current approach and no bias reduction for high-intensity events.

In addition to the work planned for a better specification of the CFIA (Section 7.4), future developments should focus on improving the current hybrid approach by accounting for the observation density observations when selecting the optimal $\beta$ values (i.e., allow for spatial variability of $\beta$). Therefore, the hybrid system could benefit places without observations while not degrading the analysis at locations with a high density of observations. This task will be facilitated by the availability of a new field informing on the amount of data assimilated (Section 7.4). Other works should explore the possibility of extending the current approach to create an ensemble version of the precipitation analysis. The REPS members, with prior bias corrections, could be used to create the analysis background field members. The hybrid approach could also help create a global version of CaPA using the GDPS and the GEPS (Houtekamer et al., 2014), respectively, as a background model and an ensemble system.



A global version of CaPA has already been identified as useful for land surface data assimilation at CCMEP. In that context, the hybrid approach would be especially relevant for the many regions of the world that do not benefit from high-density precipitation observations.

*Data availability.*  REPS and CaPA (regional configuration) are both available as operational systems of ECCC at the following locations, respectively, https://dd.meteo.gc.ca/ensemble/reps/10km/grib2 and https://dd.meteo.gc.ca/analysis/precip/rdpa/grib2/polar_stereographic. The output data from this study (CaPA using hybrid methodology for the assimilation) have been archived and are available upon request to the corresponding author.

*Author contributions.*  The main contributions from each co-author are as follows: under the supervision of SB and VF, DK proposed the
methodological, the experimental design, and the preparation of the paper. GR contributed to the adaptation of the CaPA data assimilation code to integrate the hybrid modeling. VF, SB, and FL contributed to interpreting results and paper preparation.

*Competing interests.*  The authors declare that they have no conflict of interest.





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





**Figure 1.** Blue and orange dots represent, respectively, the various precipitation networks and the SYNOP stations assimilated in summer 2019. Shaded gray areas illustrate the assimilated portion of radar beams. The top right panel illustrates the analysis domain.

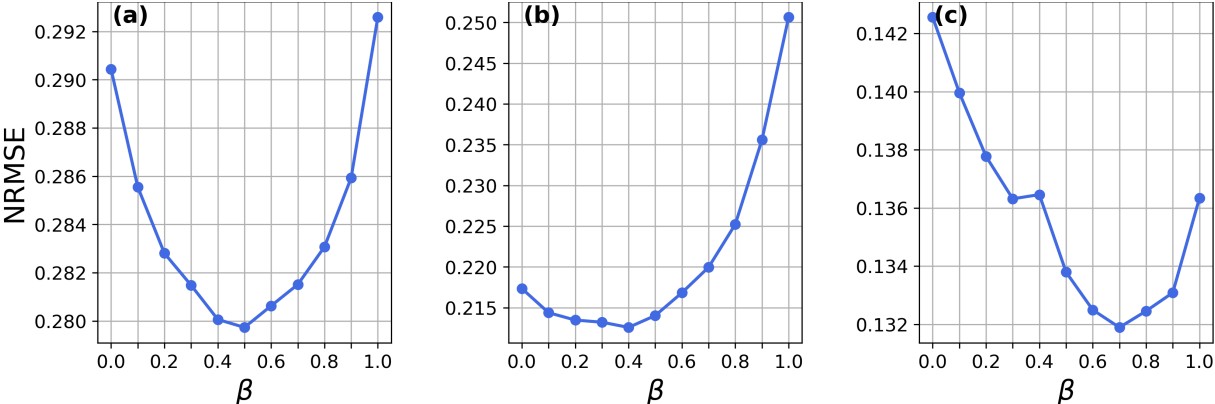

**Figure 2.** NRMSE averaged over the summer period without (a) and with (b) the assimilation of radar QPEs and over the winter (c) for the different $\beta$ values.

**Table 1.** The $2 \times 2$ Contingency table for binary events where $a$, $b$, $c$ and $d$ corresponds to the frequency of hits, false alarm, misses and correct negatives.

|  | Observed event Yes | Observed event No | Total |
|---|---|---|---|
| Analysis event Yes | a (hit) | b (false alarm) | a+b |
| Analysis event No | c (miss) | d (correct negative) | c+d |
| Total | a+c | b+d | 1 |



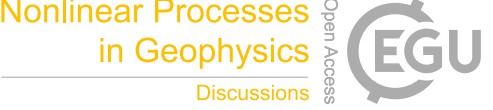

## Summary without radars

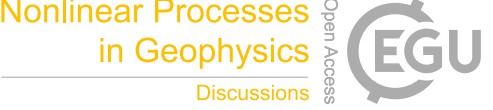

**Figure 3.** FBI-1, ETS, POD and FAR across the whole domain for summer experiment without radar QPEs for precipitation analysis with $\beta = 0.0$ (dark blue line), $\beta = 0.5$ (yellow line in a) and $\beta = 0.3$ (yellow line in b). Same figures but for the summer experiment with the assimilation radar QPEs with $\beta = 0.4$ (yellow line c) and $\beta = 0.3$ (yellow line d) both compared to the reference experiment when $\beta = 0.0$ (blue line). Filled markers indicate no significant differences at the 95% confidence level between the reference experiment $\beta = 0.0$ and $\beta = 0.5$, $\beta = 0.4$ or $\beta = 0.3$ experiments.


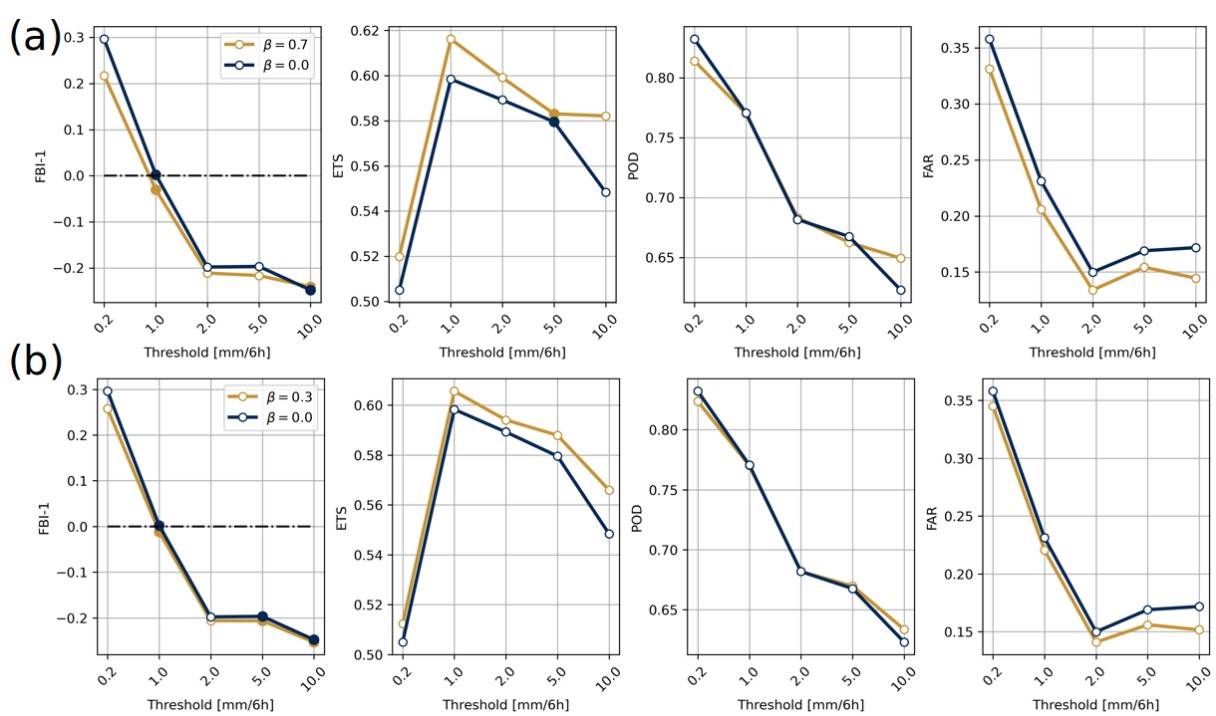

**Figure 4.** Same as Fig. 3 but for the winter experiment and with $\beta = 0.7$ (top panel) and $\beta = 0.3$ (bottom panel), both compared to the reference experiment when $\beta = 0.0$.

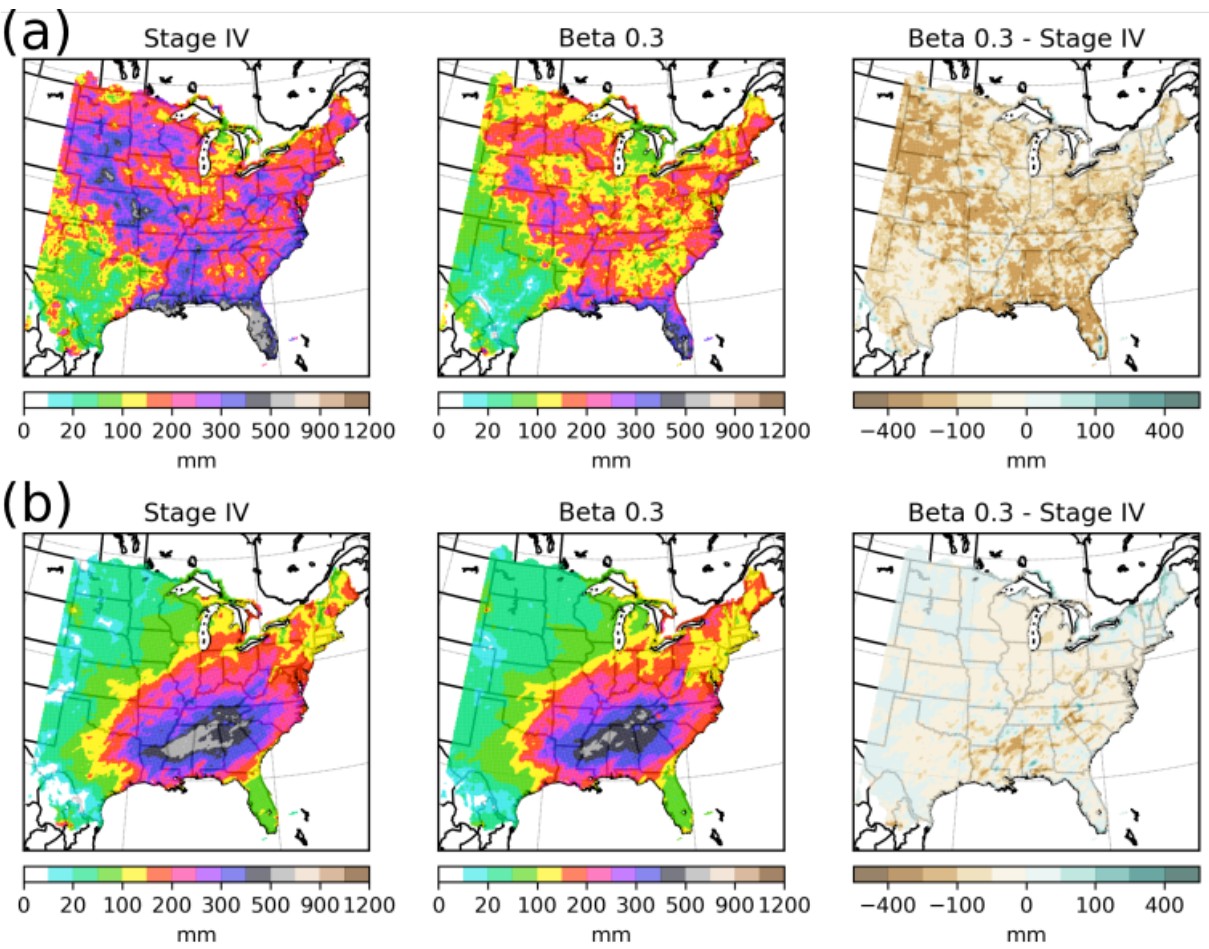

**Figure 5.** (a) From left to right, 6-h precipitation accumulated over the summer for ST4, for CaPA with $\beta = 0.3$, and the difference between the two; (b) same as (a) but for the winter experiment.



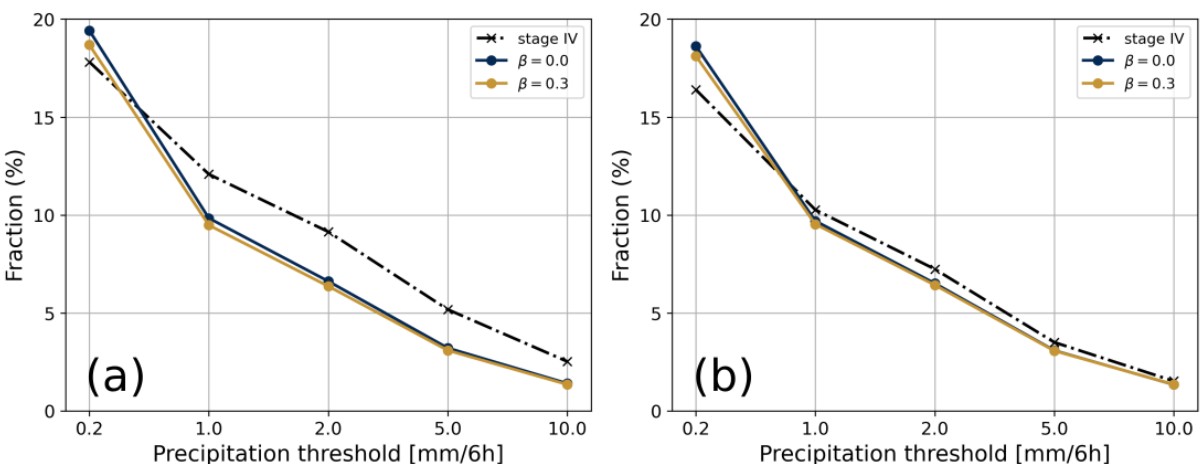

**Figure 6.** Fractional areal coverage (%) of 6-hour accumulated precipitation meeting or exceeding selected thresholds ($x$-axis) during the summer (a) and the winter (b); for ST4 (dashed black line) and CaPA using the configuration with $\beta = 0.0$ (blue line) and 0.3 (yellow line) over their common domain (Fig. 5).





**Figure 7.** Fraction skill score (FSS) as a function of the different $\beta$ values for 6-hour precipitation meeting or exceeding 0.2 mm (a,c) and 1.0 mm (b,d) thresholds for summer (first row) and winter (second row); using ST4 precipitation as reference. Each curve represents different neighboring length scale (20, 30 and 50 km).

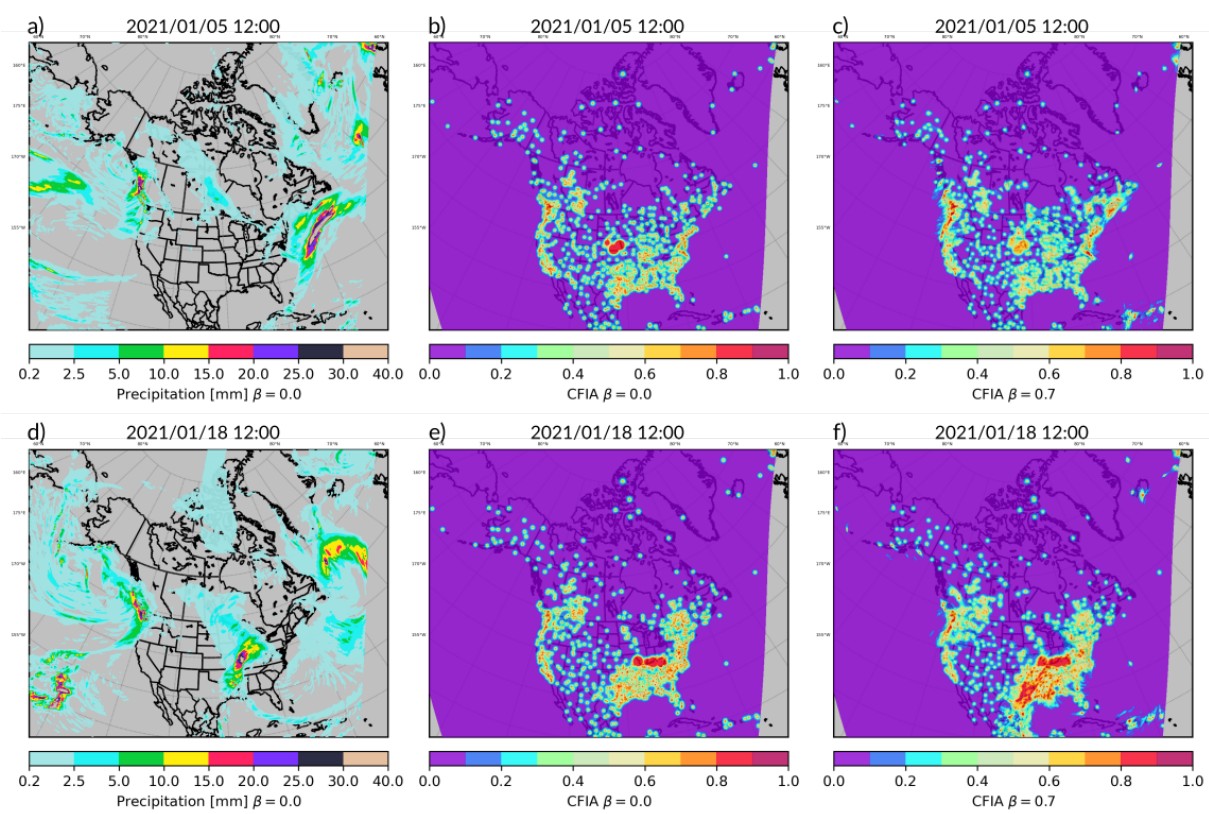

**Figure 8.** Precipitation field (a,d) from CaPA and associated CFIA values (b,e) for the 2020/01/05 12:00 and 2020/01/18 12:00 valid time. Panel (c) and (f) shows the CFIA for the same date but using the hybrid approach with high contribution of the REPS, here with $\beta = 0.7$.