# Peer review of "Using a Hybrid Optimal Interpolation-Ensemble Kalman Filter for the Canadian Precipitation Analysis"

_Nonlinear Processes in Geophysics, 2022_

## Referee Comment (RC2)

**Review manuscript number npg-2020-10: reviewer comments to the authors**

May 12, 2022

**1 Reviewer general comments to the authors**

This paper is about the use of an hybrid covariance data assimilation scheme within the Canadian Precipitation Analysis. The paper is of very good quality, very well written, nice to read, with a very rigorous presentation of the elements of the study, the observations, the model, the scores used, the results etc. Despite using a methodology that is quite familiar now, an hybrid covariance scheme, the paper is quite innovative as it presents, to my knowledge, one of the first application of an hybrid scheme to precipitation forecasts. The authors provide a very convincing demonstration of the pre-eminence of the hybrid scheme over an Optimal Interpolation scheme only. This demonstration is based on different scores like the normalized root mean square error or the scores derived from a contingency table for binary events like for example the equitable threat score or the false alarm ratio and also with the comparison against ST4 data. In particular they show that the hybrid performs better than the OI only during both winter and summer seasons and in particular confirm previous findings from [Wang et al., 2008] of the effectiveness of the hybrid scheme with sparse observations networks.

All in all I had a hard time finding anything relevant to say about that article just because it is so well written. That said, if I was to say one flaw is that the authors do not emphasize enough the fact that the hybrid scheme not only performs better than the full static case but also better than the full dynamic case. They actually talk about it only in the conclusion. I imagine that the authors are interested in the improvement of their data assimilation system compared to the OI version of it, but they must understand that the fact that the hybrid performs also better than the EnKF for precipitation forecasting can be of great interest for the rest of the community. I would suggest the authors to complete their analysis in both sections 7.1 and 7.2 by commenting further about the full dynamic case, and also to complete the figures 3 and 4 by adding the line of the case $\beta = 1$. I sincerely believe that it would help improving the paper and that it would not require too much work from the authors.

**2 Specific comments**

**Page 3, line 76**: "$\beta$ is comprised between 0 and 1, ensuring that the total background error covariances are conserved". This is true if the matrices $\mathbf{P}_{\mathrm{OI}}^{\mathrm{b}}$ and $\mathbf{P}_{\mathrm{d}}^{\mathrm{b}}$ provide "independent estimations of the true background error covariance matrix", [Ménétrier and Auligné, 2015]. So, I

would be grateful to the authors if they could go a little bit more through that point, and explain why they think that the matrix $\mathbf{P}_{\mathrm{OI}}^{\mathrm{b}}$ they build represents an estimation of the true background error covariance matrix.

**Page 4, eq. (4):** what do the notations $z_i$ and $z_j$ stand for? Is that the value of the innovations at locations $i$ and $j$? Please, add the definition of $z_i$ and $z_j$ after eq. (4).

**Page 4, line 95:** I am not familiar with variographic analysis, in my understanding, you use eq. (4) to fit it on the empirical semivariogram and determine an optimal value of $\sigma_{\mathrm{OI}}^2$. If I am correct, please can you add a sentence clarifying that point here (even though this is also specified page 5, line 119), otherwise it is unclear why you introduce this function $\gamma$ here.

**Page 4, lines 105-106:** I am aware that this comment is obvious but in order to speed-up the computation you could also perform the analysis for each grid-cell in parallel. I guess it would not require a lot of modifications to the existing version of the code. I have no idea how much the computation efficiency is critical in this case though.

**Page 7, line 185, eq. (13):** that criteria for rejecting observations is baffling to me, I feel like I missed something. Eq. (13) basically means that if the absolute difference between the Box-Cox of the observation and that of CaPA is smaller than a specific threshold then the observation is rejected. While you would like to reject observations that are too "far" from the model to avoid too strong updates. Can the authors correct that point? Or just let me know if I missed something.

**Page 7, lines 190-191:** if I am not mistaken, I have counted so far 3 quality checks, maybe it could be an idea to summarize them in a table.

**Page 7, line 199:** "seamless precipitation fields", I do not know here if this is my english that is at fault or my limited knowledge of precipitations, but I do not know what is a "seamless precipitation field", can you precise it between parenthesis maybe, or add a reference if necessary?

**Page 9, lines 256-263:** based only on the shape on the curves it seems that the hybrid approach brings potentially a dramatic improvement compared to the OI only based approach. Though, a quick calculation shows that the relative reduction of NRMSE of the hybrid approach for the optimal value of $\beta$ is rather limited with around 3.4%, 2.3%, and 7% reduction of NRMSE, respectively for fig. 2-(a), 2-(b), and 2-(c) (though I must say that in the case of winter 7% is quite good). I would then recommend the authors to go a little bit more through that in that paragraph.

Also, the authors have missed an opportunity here to deepen their analysis and show the benefits one could retrieve from the use of an hybrid scheme, not only compared to the full static case, $\beta = 0$, but also compared to the full dynamic case, $\beta = 1$. Indeed, the authors do not mention that case while at the same time they show that the hybrid performs better than the EnKF only. What I mean is that if the hybrid was performing better than the static case only but no better than the dynamic case it would be of no interest. So, despite the reference case of the authors being $\beta = 0$ I would highly recommend that they treat the case of the standalone EnKF only for the reason aforementioned and that they complete that paragraph accordingly.

**Sections 7.1 and 7.2:** the authors definitely have to talk more about the case $\beta = 1$. The authors could complete the figures 3 and 4 by adding the curve for $\beta = 1$ and then complete their analysis by emphasizing the fact that the hybrid also improves the results compared to the full dynamic case. I do believe that it would not require too much work from the authors while improving the quality of the paper.

**3   Technical corrections**

**Page 3, line 73:** repetition: "the the background field".
**Page 4, line 96:** $\mathbf{P}_{OI}^{a}$, is it an error in the notation? Should not it be $\mathbf{P}_{OI}^{b}$?
**Page 4, eq. (6):** you did not define what is $\mathbf{A}$.
**Page 5, line 126:** I would recommend not to write "(1) minus..." but "1 minus". The notation (1) is misleading and can make think about the numerotation of an equation.
**Page 6, line 155:** the acronym SYNOP is not defined, does it stand for synoptic?
**Page 9, eq. (16):** it seems that there are a few mistakes in the writing of eq. (16), I guess eq. (16) writes:

$$FSS = 1 - \frac{\frac{1}{N_y} \sum_{i=1}^{N_y} \left( f_a(i) - f_o(i) \right)^2}{\frac{1}{N_y} \left[ \sum_{i=1}^{N_y} f_a(i)^2 + \sum_{i=1}^{N_y} f_o(i)^2 \right]} \tag{1}$$

**Page 11, line 296:** repetition: "the POD slightly was slightly deteriorated".
**Page 14, line 425:** repetition: "for the observation density observations"

**References**

[Ménétrier and Auligné, 2015] Ménétrier, B. and Auligné, T. (2015). Optimized localization and hybridization to filter ensemble-based covariances. Monthly Weather Review, 143(10):3931–3947.

[Wang et al., 2008] Wang, X., Barker, D. M., Snyder, C., and Hamill, T. M. (2008). A Hybrid ETKF − 3DVAR Data Assimilation Scheme for the WRF Model . Part I : Observing System Simulation Experiment. (Lorenc 2003):5116–5131.

---

## Author Comment (AC1)

**Response to reviewer #1**

Dear Anonymous Referee #1,

We want to thank referee #1 for the review and the opportunity to improve our paper. We hope we have adequately answered all the reviewer' comments addressed in the following with a point-by-point response in *italic*. Sentences that we suggest for addition or modification to the revised version of the manuscript are indicated in *italic blue*.
Best regards,

Dikraa Khedhaouiria on the behalf of all co-authors

**Highlights**

The paper presents an extension of the classical 'static' data assimilation to incorporate ensemble forcast and by such allowing to releveive some restricting constraints on the shape of the covariance matrix of the errors terms in the classical DA approach. This approach is interesting, and make use nicely of some recent developments in DA.

**General comments**

The paper is rather easy to read and is well structured, altough I got quite lost in all the acrynonims, maybe a list of them could be appreciated.

- *We thank the reviewer for this positive comment. About the acronyms, we suggest an Appendix that will list all the acronyms present in the manuscript (see page 10 of this document for the appendix).*

As I am not directly an expert of the domain, I do not know what are the models, so it took me some time to figure out what how it is constructed. It may worth introducing the whole model in section 2 (analysis and observations models).

- *We think that it is essential that non-expert in data assimilation approaches can follow the thread of the article. For this reason, we intend to restructure the article so that current Section 4, which describes the datasets, to be inserted as section 2. The current Section 2, which presents the assimilation approaches, will be Section 3, and so on for the remaining sections. However, as the observation's quality control described in lines 176-195 needs some aspects of the assimilation to be defined first, it will be moved as a subsection of the new Section 3. Therefore, we propose the following structure for the next version of the manuscript:*

  - *2. Datasets*
  - *2.1 Model Description*
  - *2.2 Observations*
  - *2.3 Stage IV precipitation*
  - *3. Methodology*
  - *3.1 Hybrid assimilation approach for the precipitation analysis*
  - *3.2 Quality Control of the observations during the assimilation*

  *Except for the section number, the other sections will not be restructured. Finally, it is worth mentioning that with this new plan, small rearrangements of the reviewed manuscript were needed:*

  - *Lines 68-71 currently as: "Section 2 introduces the hybrid analysis scheme, while Section 3 describes the method to select the optimal weighting for the hybrid approach. Section 4 and 5 present respectively the datasets and experimental design followed by the verification strategy in Section 6. The results of the experiments are made available in Section 7 and the conclusion and the discussions are given in Section 8." will be changed for "Sections 2 and 3 introduce the datasets and the hybrid analysis scheme methodology, respectively. Section 4 describes the method to select the optimal weighting for the hybrid approach, while section 5 presents the experimental design followed by the verification strategy in Section 6. The results of the experiments are made available in Section 7, and the conclusion and the discussions are given in Section 8."*

– *Reference to LOO in lines 197-198 will be removed as it is not yet introduced at this point of the proposed version of the manuscript. Lines 197-198 currently as: "In addition to the LOO verification at station locations, the 6-hour analyses were also compared to 6-h Stage IV (ST4) analysis from the National Centers for Environmental Prediction (NCEP, Lin and Mitchell, 2005)."* will be changed for *"In addition to verification at station locations (see Section 3.2 below), the 6-hour analyses were also compared to 6-h Stage IV (ST4) analysis from the National Centers for Environmental Prediction (NCEP, Lin and Mitchell, 2005)"*.

– *As $\beta$ was also not yet introduced in the proposed Section 2.3, the sentence in lines 205-206 "However, ST4 is a valuable dataset that could help compare the results obtained with the different $\beta$ values in the hybrid approach."* will be modified such as: *"However, ST4 is a valuable dataset that could help compare the results obtained with the different configurations of the hybrid approach."*

In Eq 6, the authors use the approach of Hamill and Snyder to estimate the variance of the background errors, but it seems that they do not do the same for the hydrid approach (Eq 10), I may be wrong, or may the authors comment on that ?

- *We apologize for the unclear explanation, and we thank the reviewer for this comment for two reasons. First, it helped us catch errors in equations (5) and (10). The denominator should have been $N-2$ instead of $N-1$ (see equation 5 in Hamill and Snyder 2000). It is essential to mention that this typo is not present in the analysis computation; therefore, no changes are expected in the results. Equation (5) will be corrected for:*

$$P^b{}_d = \frac{1}{N-2} A' A'^T$$

*Second, we did use in equation (10) the anomaly computation as explained in equation (6)[Hamill and Snyder 2000], and we agree that the way we explained the meaning of $\overline{x}_{s_o}$ can be misleading. For these reasons, we intend to change the following (line 120-121 p.5) "[...]:*

$$\sigma^2_{b,REPS}(s_o) = \frac{1}{N-1} \left( x^{(REPS)}_{s_o} - \overline{x}_{s_o} \right) \left( x^{(REPS)}_{s_o} - \overline{x}_{s_o} \right)^T, \tag{1}$$

*where $x^{(REPS)}_{s_o}$ is the N-length vector of REPS precipitation at location $s_o$ and $\overline{x}_{s_o}$ is the ensemble mean at the same location." for "[...]:*

$$\sigma^2_{b,REPS}(s_o) = \frac{1}{N-2} \sum_{i=1}^{N} \left( x^{(REPS)}_{s_o,i} - \overline{x}_{s_o[\![1,N]\!]\backslash i} \right) \left( x^{(REPS)}_{s_o} - \overline{x}_{s_o[\![1,N]\!]\backslash i} \right)^T, \tag{2}$$

*where $x^{(REPS)}_{s_o}$ is the N-length vector of REPS precipitation at location $s_o$ and $\overline{x}_{s_o[\![1,N]\!]\backslash i}$ is the ensemble mean at the same location as defined in equation (5)."*

Concerning the results I am a bit surprised that the performance curve un beta (figure 2-c) goes up between .3 and .4, is a sampling artifact ? Maybe the authors should add some comment on this fact or provide some estimation of the variability of the NRMSE.

- *Figure 2.c illustrates the NRMSE values that help compare the analysis, for different $\beta$s, during the winter against observation at surface stations in a leave-one-framework. The reviewer is right to mention the peculiarities present for $\beta = 0.3$ and $0.4$. We carefully checked, and the difference between the two NRMSEs is around $0.1\%$ and is indeed a sampling effect.*

In the same objective of better understanding the gain linked to the assimilation of the data in the model, would it be possible to compute the metrics (FBI-I, ETS, POD and FAR) when $\beta$=1 ?

- *We thank reviewer #1 for this comment also raised by reviewer #2. We will comment and add results when $\beta$=1 at different places in the new version of the manuscript. The following items list all the modification suggestions:*

  - *Figures 3 and 4 will additionally display the metric (FBI-I, ETS, POD, FAR) values when $\beta = 1$ (grey curve). The new version of these figures and their captions are provided at the end of this document (pages 11-12). We want to draw the reviewer's attention to Figure 3.d. An error occurred while merging different figures. In the current manuscript version, Figure 3.d is the same as Figure 4.b, which is wrong. We will correct Figure 3 accordingly. As shown on page 11 of this document, the results are much more consistent with the obtained NRMSE values (Figure 2.b). Indeed, using $\beta = 0.3$ and $\beta = 0.4$ during summer, with the assimilation of radar QPEs, lead to similar outputs in verification metrics.*

[revised manuscript text omitted]

In the metrics, the authors point out the values that are not significantly different, maybe they could also plot the variability (errors bars, or boxplots ?)

- *We thank the reviewer for this interesting comment. The statistical significance of the difference between scores of different configurations regards Figures 3 and 4, for a total of 24 subplots. In the revised version of the manuscript, each subplot will contain the results for $\beta = 1.0$ (see reply to the previous comment), leading to three curves, and adding the variability information for each of this curve would, therefore, highly burden Figure 3 and 4, without adding much information. We hope it is okay if we do not make any modifications regarding the variability in the scores.*

**Specific issues**

- Eq 1, the model could be presented first, so that we know what $\widetilde{\mathrm{P}}^b$ corresponds to. In particular, it could be useful to have the size of the matrices

- *The introduction of the forecasting models before presenting the hybrid approach will be addressed in the revised version of the manuscript. This topic is thoroughly discussed in the General Comments section above. $\widetilde{P}^b$ is a covariance matrix and is therefore by definition a square matrix between each pair of elements of a given random vector. In the context of data assimilation, the covariance matrix of the background error is computed at each grid-cell of the domain using the neighboring grid-cells where observations are collocated. If there are $M$ locations with observations in the vicinity of a grid-cell, the background values (i.e., the REPS and the RDPS) at those locations will be used to build a $M \times M$ covariance matrix. In CaPA, no matter the configuration, we use a maximum of 16 points per type of observation in the vicinity when building covariance matrices. Past testing experiments led to this choice and helped speed up the analysis computation while ensuring a good quality product. Moreover, the matrice size is already discussed in Lines 105-109. We added information on the matrix size when confusion was possible, for example, in equations 5 and 6. For these reasons, we suggest not adding more information regarding the size of the different matrices, and we hope the reviewer will understand this point.*

- Eq 5 the prime notation is introduced a bit too early I guess, and $P_{OI}^{\alpha}$ (L96) is not yet defined. Similarly, $A_{:,j}$ (eq 6) is not defined.

- *We are sorry for the confusion here. We will change the $P_{OI}^a$ for $P_{OI}^b$. $P_{OI}^a$ corresponds to the analysis error covariance matrix when the analysis is solely based on the OI approach. However, it is not explicitly defined in the manuscript. We intend to keep the prime notation for the anomaly (perturbation) matrix as in equation (5) as it is not the conventional way to compute it. However, the reviewer is correct about the lack of clarity in Equation 5, which also has an error (see our previous comments in the General comments section). Indeed, the denominator should be $(N-2)$ and not $(N-1)$. For these reasons, we propose to change the following:*

    - **line 96-104**: *"The covariance matrix $P^b{}_d$ depicts the flow-dependant errors estimated from the REPS and is defined as:*

$$P^b{}_d = \frac{1}{N-1} A' A'^{T}, \tag{3}$$

*where $A' \in R^{m \times N}$ denotes the anomalies estimated from the $N$-member ensemble at $m$ grid points. The superscript $T$ corresponds to a matrix transpose. To avoid underestimation of the variance of the background errors due to the limited size of the dynamical ensemble (Houtekamer and Mitchell, 1998), the anomaly computation follows Hamill and Snyder (2000) suggestions and writes as:*

$$A'_{:,j} = A_{:,j} - \overline{A}_{[\![1,N]\!] \setminus j}, \tag{4}$$

*where $A'_{:,j}$ is the $j$-th column of $A'$ and $\overline{A}_{[\![1,N]\!] \setminus j}$ is the average across the $N$ members without the $j$-th member."*

    - *by: "The covariance matrix $P^b{}_d$ depicts the flow-dependant errors estimated from the REPS and is defined as:*

$$P^b{}_d = \frac{1}{N-2} A' A'^{T}, \tag{5}$$

*where $A' \in R^{m \times N}$ denotes the anomalies estimated from the $N$-member ensemble at $m$ grid points. The superscript $T$ corresponds to a matrix transpose. To avoid underestimation of the variance of the background errors due to the limited size of the dynamical ensemble (Houtekamer and Mitchell, 1998), the anomaly computation follows Hamill and Snyder (2000) suggestions and writes as:*

$$A'_{[:,j]} = X_{f[:,j]} - \overline{X_f}_{[\![1,N]\!]\setminus j},\tag{6}$$

*where the subscript $[:,j]$ refers to the $j$-th column of a given matrix and $X_f$ is $(m \times N)$ precipitation ensemble forecast matrix. The $m$ lengh vector $\overline{X_f}_{[\![1,N]\!]\setminus j}$ corresponds to the average across the $N$ members without the $j$-th member."*

- L155: I understand that SYNOP is a netwrok of stations, but the term is not introduced earlier.

- *The reviewer is totally right, SYNOP stands for synoptic stations and the acronym was not defined. We will modify*

  – **line 155**: *"For more reliability, the NRMSE is computed only at SYNOP and manual SYNOP stations during the summer and [...]"*

  – *by: "For more reliability, the NRMSE is computed only at surface synoptic observations (hereafter SYNOP) and manual SYNOP stations during the summer and [...]"*

  *Moreover, the suggested Appendix A (see page of the present document) will also define SYNOP among other acronyms.*

- L174: 'progressively' if quite unprecise, if the authors think it is worth mentioning, I think it should be precised;

- *We are sorry for the lack of precision on this point. The sentence in line 174 : "New Canadian dual-polarization Doppler radars have been progressively added to the observation database." refers to the ongoing replacement of C-band to S-band radars initiated in 2017 (see $https://www.canada.ca/en/environment-climate-change/services/weather-general-tools-resources/radar-overview/modernizing-network.html$). We propose to change line 174 for "Since 2017, new Canadian dual-polarization Doppler radars have been progressively replacing their C-band counterpart."*

- L187: 'in the transformed space', does it refer to the box-cox transformed data ?

- *Yes, it does. We use this comment to point out that, thanks to reviewer #2, an observation is rejected if eq. (13) is **not** satisfied and not the other way around as specified in the current version of the manuscript. Therefore, we propose the following modification to help better understand the CaPA quality control and correct the misinterpretation of eq. 13. :*

  – *line 183-187: "For this purpose, an analysis is estimated at a site $s_k$ using neighboring stations in a LOO approach. The observation is rejected or, is said invalid, if:*

  $$\left| x_{s_k}^{(OBS)} - x_{s_k}^{(CaPA)} \right| < tol \cdot \sqrt{(\sigma_o^2 + \sigma_a^2)},\tag{13}$$

  *where tol is a tolerance factor set equal to 4 for the operational CaPA, and $obs$ and $CaPA$ superscripts refer to, respectively, the observation and the analysis in the transformed space"*

- by: *For this purpose, an analysis is estimated at a site $s_k$ ($x_{s_k}^{(a)}$) using neighboring stations in a LOO approach. The observation at the same site ($x_{s_k}^{(o)}$) is rejected or, is said invalid, if the following is not fulfilled:*

$$\left| x_{s_k}^{(o)} - x_{s_k}^{(a)} \right| < tol \cdot \sqrt{(\sigma_o^2 + \sigma_a^2)}, \tag{13}$$

*where tol is a tolerance factor set equal to 4 for the operational CaPA. Both observed precipitation and analysis estimates of equation (13) undergone the Box-Cox transformation beforehand (see eq. 2).*

- L201: what are the 'robustness reasons' ?

  • *We are sorry to have taken the reader's knowledge of Stage IV product performance for granted. Stage IV results from an advanced blending of surface stations and radar datasets, but this product still suffers from bias and has some limitations, especially in the western CONUS domain. Nelson et al. (2016) study provides a thorough picture of the strengths and limitations of Stage IV. The lack of reliability of radar data in complex terrain is one reason to discard the western CONUS domain. It is a common practice when Stage IV is used as a reference for verifications. A quick search in Google Scholar for papers quoting Nelson et al. (2016) article provides an idea of how common it is to not use the western part of the CONUS domain in Stage IV. We, therefore, suggest replacing the following:*

    - lines **200-202**: *"The ST4 domain covers the contiguous United States (CONUS), but for robustness reasons, only the CONUS east of 105W was used for verification purposes [...]"*
    - by: *"The ST4 domain covers the contiguous United States (CONUS), but for known limitations in the western domain (see Nelson et al. 2016 for more details), only the CONUS east of 105W was used for verification purposes [...]"*

- Eq 16: the sum in the denominator are between 1 and $N_y$ ?

  • *We thank the reviewer for catching these typos. Reviewer #2 also highlighted that a bracket is not at the right place in the numerator. Therefore, we will modify equation 16, initially written as:*

    - **line 246**: *"*

$$FSS = 1 - \frac{\frac{1}{N_y} \sum_{i=1}^{N_y} \left( f_a(i) - f_{o(i)} \right)^2}{\frac{1}{N_y} \left[ \sum_{i=1}^{n} f_{a(i)}^2 + \sum_{i=1}^{n} f_{o(i)}^2 \right]} \tag{16}$$

    *"*

    - for *"*

$$FSS = 1 - \frac{\frac{1}{N_y} \sum_{i=1}^{N_y} \left( f_{a(i)} - f_{o(i)} \right)^2}{\frac{1}{N_y} \left[ \sum_{i=1}^{N_y} f_{a(i)}^2 + \sum_{i=1}^{N_y} f_{o(i)}^2 \right]} \tag{16}$$

    *"*

- L374: the "two different interpretations" should be precised, they are not clear (at least to me)

  • *We are sorry for not being precise enough about the difficulty of interpreting low CFIA values when using the hybrid approach. Section 7.4 discusses the limitations of using the CFIA field in its definition (eq. 11). Presently, for a given grid-cell, if there are no stations or radars in the vicinity, the analysis is set equal to the background field (lines 109-110), therefore in equation 11, $\sigma_a^2$ will tend to be equal to $\sigma_b^2$, and CFIA will tend towards low values. This way of computing CFIA is*

true no matter the value of $\beta$. Increasing the contribution of the REPS, i.e., increasing $\beta$, leads to CFIAs following meteorological spatial distributions, which is a nice feature. However, when no precipitation is observed in ensemble forecasts, the anomaly matrix tends towards a null matrix. Thus the matrix $P_d^b$ (eq. 5), tends towards zero. Here again, $\sigma_a^2$ as defined in equation 9 will tend to be equal to $\sigma_b^2$ and again, CFIA values will be small. Therefore, the user may ask if the small CFIA values he/she is reading reflect the absence of precipitation or the absence of assimilated observations. To better illustrate this point, we intend to add after line 369:

– "Generally, CFIAs tended to be higher at places with precipitation than when using $\beta = 0.0$, as shown in the eastern part of the domain for the January 18 case (Fig 8.f). Inversely, locations with no precipitation in the background field tended to show small CFIA values. This last result is consistent with the calculation of $P_d^b$ (eq. 5), which, when the ensemble members have no or very little precipitation, will tend to a zero matrix."

We also suggest to modify line 370-371 (p.13):

– "This is explained by the current CaPA computation framework, where gridcells are set equal to the background when no observations are available in the vicinity."

– for "'This is explained by the current CaPA computation framework. No matter the value of $\beta$, analysis precipitation at a given grid-cell is equal to the background when no observations to be assimilated are available in the vicinity. The latter lead to error variances of the analysis ($\sigma_a^2$) being close to those in the background ($\sigma_b^2$), and by construction CFIA will tend towards small values (eq. 11)."

**Appendix A. List of acronyms**

| | |
|---|---|
| CaPA | Canadian Precipitation Analysis |
| CFIA | ConFidence Index of the Analysis |
| DA | Data Assimilation |
| ECCC | Environement and Climate Change Canada |
| EnKF | Ensemble Kalman Filter |
| EPS | Ensemble Prediction System |
| GEPS | Global Ensemble Forecast System |
| ETS | Equitable Threat Score |
| FAR | False Alarm Rate |
| FBI | Frequency Bias Index |
| FSS | Fraction Skill Score |
| LOO | Leave-one-out |
| NRMSE | Normalized Root Mean Squared Error |
| NWP | Numerical Weather Prediction |
| OI | Optimal Interpolation |
| POD | Probability of Detection |
| QC | Quality Control |
| QPE | Quantitative Precipitation Estimates |
| RDPS | Regional Deterministic Prediction System |
| REPS | Regional Ensemble Prediction System |
| ST4 | Stage IV |
| SYNOP | Synoptic stations |

**Summary without radars**

[Figure]

**Summer with radars**

[Figure]

**Figure 3**. FBI-1, ETS, POD and FAR across the whole domain for summer experiment without radar QPEs for precipitation analysis with $\beta = 0.0$ (dark blue line), $\beta = 1.0$ (grey line), $\beta = 0.5$ (yellow line in a) and $\beta = 0.3$ (yellow line in b). Same figures but for the summer experiment with the assimilation radar QPEs with $\beta = 1.0$ (grey line), $\beta = 0.4$ (yellow line c) and $\beta = 0.3$ (yellow line d) all three compared to the reference experiment when $\beta = 0.0$ (blue line). Filled markers indicate no significant differences at the 95% confidence level between the reference experiment $\beta = 0.0$ and $\beta = 1.0$, $\beta = 0.5$, $\beta = 0.4$ or $\beta = 0.3$ experiments.

[Figure]

**Figure 4.** Same as Fig. 3 but for the winter experiment and with $\beta = 0.7$ (top panel), $\beta = 0.3$ (bottom panel) and $\beta = 1.0$ (two panels), all three compared to the reference experiment ($\beta = 0.0$).

---

## Author Comment (AC2)

**Response to reviewer #2**

Dear Anonymous Referee #2,

We want to thank referee #2 for the review and the opportunity to improve our paper. We hope we have adequately answered all the reviewer's comments. Reviewer comments are addressed in the following with a point-by-point response in *italic*. Sentences that we suggest for addition or modification to the revised version of the manuscript are indicated in *italic blue*.

Best regards,

Dikraa Khedhaouiria on the behalf of all co-authors

**1. Reviewer general comments to the authors**

This paper is about the use of an hybrid covariance data assimilation scheme within the Canadian Precipitation Analysis. The paper is of very good quality, very well written, nice to read, with a very rigorous presentation of the elements of the study, the observations, the model, the scores used, the results etc. Despite using a methodology that is quite familiar now, an hybrid covariance scheme, the paper is quite innovative as it presents, to my knowledge, one of the first application of an hybrid scheme to precipitation forecasts. The authors provide a very convincing demonstration of the preeminence of the hybrid scheme over an Optimal Interpolation scheme only. This demonstration is based on different scores like the normalized root mean square error or the scores derived from a contingency table for binary events like for example the equitable threat score or the false alarm ratio and also with the comparison against ST4 data. In particular they show that the hybrid performs better than the OI only during both winter and summer seasons and in particular confirm previous findings from [Wang et al., 2008] of the effectiveness of the hybrid scheme with sparse observations networks.

All in all I had a hard time finding anything relevant to say about that article just because it is so well written. That said, if I was to say one flaw is that the authors do not emphasize enough the fact that the hybrid scheme not only performs better than the full static case but also better than the full dynamic case. They actually talk about it only in the conclusion. I imagine that the authors are interested in the improvement of their data assimilation system compared to the OI version of it, but they must understand that the fact that the hybrid performs also better than the EnKF for precipitation forecasting can be of great interest for the rest of the community. I would suggest the authors to complete their analysis in both sections 7.1 and 7.2 by commenting further about the full dynamic case, and also to complete the figures 3 and 4 by adding the line of the case $\beta = 1$. I sincerely believe that it would help improving the paper and that it would not require too much work from the authors.

- *Thank you very much for this positive evaluation of the manuscript. We agree that adding information about the impact on scores when using $\beta$=1.0 would give a better overall picture of the methodology. The fact that reviewer #1 also raised this concern prompts us to propose changes in this direction to our manuscripts. The last comment in Section 2 below provides a more thorough response to this concern.*

**2. Specific comments**

**Page 3, line 76**: "$\beta$ is comprised between 0 and 1, ensuring that the total background error covariances are conserved". This is true if the matrices $\mathbf{P}_{\mathrm{OI}}^{\mathrm{b}}$ and $\mathbf{P}_{\mathrm{d}}^{\mathrm{b}}$ provide "independent estimations of the true background error covariance matrix", [Ménétrier and Auligné, 2015]. So, I would be grateful to the authors if they could go a little bit more through that point, and explain why they think that the matrix $\mathbf{P}_{\mathrm{OI}}^{\mathrm{b}}$ they build represents an estimation of the true background error covariance matrix.

- *We thank the reviewer for this interesting comment. Modeling the error covariance matrices (background and observations) is a crucial step in data assimilation and is a continuous research area (Cheng et al., 2019).*

  *Even though they do not necessarily agree on this assertion, Ménétrier and Auligné (2015) explained that parameters used to weight the ensemble-based ($P_e$) and the static covariance matrices ($P_c$) sum to one (which is our case: $\beta + [1 - \beta] = 1$) because of the assumption that $P_e$ and $P_c$ provide independent estimates of the true background error covariance matrix. If we understand the comment well, reviewer #2 has no problem assuming that $P_e$ is a good estimate of the forecast error covariance as it is very well documented (see, for example, Lorenc 2003). It is indeed one of the major hypotheses in the Ensemble Kalman Filter. Reviewer #2 is therefore questioning how*

*our $P_c$ matrix, i.e., $\mathbf{P}^b_{OI}$, is also an estimate of the true background error covariance matrix. The answer to the latter point is that based on several assumptions and past experiments, the elements of $\mathbf{P}^b_{OI}$ matrix were supposed to be isotropic, homogeneous and follow and exponential decay (see Equation 3). This error covariance matrix structure has been successful for CaPA (see Fortin et al., 2018 for an extensive review) and other studies conducted on different variables and using such an approach (Mitchell et al. 1990, Garand and and Grassotti 1995, Brasnett 1999; among others). Although $\mathbf{P}^b_{OI}$ could be improved by relaxing some of the assumptions, this work is beyond the scope of this study. It is also interesting to note that when comparing the experiments with $\beta = 0.0$ to $beta = 1.0$ (which could be considered somehow as an EnKF configuration), results showed that $\mathbf{P}^b_{OI}$ are not completely different and displayed similarities for some precipitation thresholds. All this point, therefore, suggests that $\mathbf{P}^b_{OI}$ provides relevant information regarding the horizontal correlations of the background errors and represents an estimation of the true background error covariance matrix.*

**Page 4, eq. (4)**: what do the notations $z_i$ and $z_j$ stand for? Is that the value of the innovations at locations $i$ and $j$? Please, add the definition of $z_i$ and $z_j$ after eq. (4).

- *In line 89, page 3, we introduced both the locations $i$ and $j$ and the variable $Z$ such as: "[...] the Euclidean distance between locations $i$ and $j$ and the correlation length. These parameters are estimated using variographic analysis of the innovations, $Z = d - Hx_f$. The $d$, $x_f$ and $H$ correspond respectively to the measurements [...] ", suggesting that $z_i$ and $z_j$ would be the value of the innovations at locations $i$ and $j$. However, to add clarity, we suggest to replace line 95:*

    - *"where $\sigma^2_o$ corresponds to variance errors of the observation."*
    - *by "where $z_i$, $z_j$ and $\sigma^2_o$ correspond, respectively to the innovations at location $i$ and $j$ and to variance errors of the observation."*

**Page 4, line 95**: I am not familiar with variographic analysis, in my understanding, you use eq. (4) to fit it on the empirical semivariogram and determine an optimal value of $\sigma^2_{OI}$. If I am correct, please can you add a sentence clarifying that point here (even though this is also specified page 5, line 119), otherwise it is unclear why you introduce this function here.

- *Yes indeed, the variographic analysis is done to estimate the parameter $\sigma^2_{OI}$, but also $\sigma^2_o$ and $l_{OI}$ and is realized before the analysis as such. We mentioned in line 89, page 3, that when referring to $\sigma^2_{OI}$ and $l_{OI}$ "[...]. These parameters are estimated using variographic analysis of the innovations, $Z = d - Hx_f$.[...] ". In light of the reviewer's comment and to add further clarification, we will modify this sentence as follows: "In CaPA algorithm, parameters $\sigma^2_{OI}$, $l_{OI}$ but also $\sigma^2_o$, which is the variance errors of the observation needed to build the observation error covariance matrix (explained further below), are estimated before the analysis as such and by using a variographic analysis of the innovations. The innovations are classicaly defined as $Z = d - Hx_f$, where $d$, $x_f$ and $H$ correspond respectively to the measurements the forecasts and the observation operator, which is here the nearest neighbour interpolation (Fortin et al., 2015). [...]".*

**Page 4, lines 105-106**: I am aware that this comment is obvious but in order to speed-up the computation you could also perform the analysis for each grid-cell in parallel. I guess it would not require a lot of modifications to the existing version of the code. I have no idea how much the computation efficiency is critical in this case though.

- *The reviewer is quite right. Reducing computation time by parallel modelling in this part of the CaPA code has already been discussed internally. However, as the reviewer guessed, computation time is not a limiting factor for CaPA, at least for the moment. Some actions are already in place in*

the code to reduce computation time. For example, the covariance matrices have a maximum size of $16 \times 16$ per observation type, which leads to a maximum of $32 \times 32$, when using both surface stations and radar QPEs (see lines 105-109 of the manuscript). This size limitation allows to speed up the resolution of the matrix systems. Finally, changing this part of the code would require a non-negligible amount of work and an intensive internal process on the part of the ECCC to ensure that CaPA, as an operational system, will not suffer any degradation due to these changes. For these reasons, there are no plans to implement this change.

**Page 7, line 185, eq. (13):** that criteria for rejecting observations is baffling to me, I feel like I missed something. Eq. (13) basically means that if the absolute difference between the Box-Cox of the observation and that of CaPA is smaller than a specific threshold then the observation is rejected. While you would like to reject observations that are too "far" from the model to avoid too strong updates. Can the authors correct that point? Or just let me know if I missed something.

- *We thank the reviewer for pointing out this; the manuscript has an error when describing equation 13. This quality control aims to reject observations from stations that appear very different from the closest ones (Lespinas et al. 2015). We take advantage of this comment to also highlight that we intend to simplify the superscripts in equation 13. Therefore, we intend to correct lines 184-185 currently as:*

  - *"For this purpose, an analysis is estimated at a site $s_k$ using neighboring stations in a LOO approach. The observation is rejected or, is said invalid, if:*

    $$\left| x_{s_k}^{(OBS)} - x_{s_k}^{(CaPA)} \right| < tol \cdot \sqrt{(\sigma_o^2 + \sigma_a^2)}, \tag{13}$$

    *"*

  - *for:* *"For this purpose, an analysis is estimated at a site $s_k$ $(x_{s_k}^{(a)})$ using neighboring stations in a LOO approach. The observation at the same site $(x_{s_k}^{(o)})$ is rejected or, is said invalid, if the following is not fulfilled:*

    $$\left| x_{s_k}^{(o)} - x_{s_k}^{(a)} \right| < tol \cdot \sqrt{(\sigma_o^2 + \sigma_a^2)}, \tag{13}$$

    *"*

**Page 7, lines 190-191**: if I am not mistaken, I have counted so far 3 quality checks, maybe it could be an idea to summarize them in a table.

- *We thank the reviewer for this good suggestion. In reality, it exists several other quality control steps of input datasets before their assimilation in CaPA that were not described in the manuscript. For example, precipitation at radar pixels is cleaned using cloud cover from GOES (Geostationary Operational Environmental Satellite) images, among many other quality control checks. Detailing all the QC steps is beyond the scope of this manuscript; some of them are already well documented in Lespinas et al. (2015) and Fortin et al. (2015), as mentioned in the manuscript on page 6, line 178. Therefore, adding a table with the three quality checks (QC) might mislead the reader into inferring that the table is an exhaustive QC list. The reviewer's comments made us think that we need to reformulate the way we list the QC steps in the manuscript and emphasize that other quality control steps exist. To do so:*

  - *we suggest to change:*
    * *"A first temporal QC is performed" (p.6, L. 178) for* *"A temporal QC is performed"*

           ∗ *"A second quality control" (p.7, L. 182) for "A different quality control"*

     – *we also intend to add: "Several other QC are applied to precipitation input datasets, but their extensive description is beyond the scope of this manuscript (see Lespinas et al. 2015 and Fortin et al. 2015 for further information)."*

**Page 7, lines 199**: "seamless precipitation fields", I do not know here if this is my english that is at fault or my limited knowledge of precipitations, but I do not know what is a "seamless precipitation field", can you precise it between parenthesis maybe, or add a reference if necessary?

- *The term "seamless precipitation fields" refers to a field without discontinuities as opposed to scattered precipitation values observed at surface stations. In the manuscript, the term seamless was chosen to emphasize that Stage IV data is continuous in space and would allow different types of verification. "Seamless" is a pretty common adjective when describing spatially continuous precipitations fields, as shown by a Google search of the term "seamless precipitation fields". We suggest to change "seamless precipitation fields" to "spatially-continuous precipitation fields".*

**Page 9, lines 256-263:** based only on the shape on the curves it seems that the hybrid approach brings potentially a dramatic improvement compared to the OI only based approach. Though, a quick calculation shows that the relative reduction of NRMSE of the hybrid approach for the optimal value of $\beta$ is rather limited with around 3.4%, 2.3%, and 7% reduction of NRMSE, respectively for fig. 2-(a), 2-(b), and 2-(c) (though I must say that in the case of winter 7% is quite good). I would then recommend the authors to go a little bit more through that in that paragraph.

- *We thank the reviewer for this suggestion. It is true that quantifying the improvements and degrations in NRMSE values relative to the reference experiment would provide a better perspective of the results. We suggest to modify lines 258-265:*

    – *"NRMSE values for $\beta$ ranging from 0.0 to 0.4 were indeed very similar for the summer experiment assimilating radar QPEs (Fig 2.b), with a minimum obtained at 0.4. On the other hand, the experiment without radar QPEs showed larger variability in NRMSEs for the different $\beta$s with a minimum obtained at 0.5 (Fig 2.a). These results suggest that when the density of assimilated observations is lower, the hybrid approach brings more added value and is thus consistent with the literature (Hamill and Snyder, 2000; Wang et al., 2008a). During the summer, the use of $\beta > 0.6$ (0.9) with (without) radar assimilation deteriorated the NRMSE values compared to the reference analysis ($\beta = 0.0$).*
     *Interestingly, the winter experiments illustrated a different pattern. According to the NRMSEs (Fig 2.c), the analysis improved when the $\beta$ increased and reached a minimum at $\beta = 0.7$."*

    – *for "NRMSE values for $\beta$ ranging from 0.0 to 0.4 were very similar for the summer experiment assimilating radar QPEs (Fig 2.b), where the optimal $\beta$ equal to 0.4 corresponds to an NRMSE reduction of 2.3% compared to $\beta = 0.0$. On the other hand, the experiment without radar QPEs showed more significant variability in NRMSEs for the different $\beta$s. The optimal value of 0.5 reduces the NRMSEs by 3.4% when compared to the experiment using $\beta$=0.0. These results suggest that when the density of assimilated observations is lower, the hybrid approach brings more added value and is thus consistent with the literature (Hamill and Snyder, 2000; Wang et al., 2008a). During the summer, the use of $\beta > 0.6$ (0.9) with (without) radar assimilation reduced the NRMSE values and therefore deteriorated the analyses compared to the reference analysis ($\beta = 0.0$).*
     *Interestingly, the winter experiments illustrated a different pattern. According to the NRMSEs (Fig 2. c), the analysis improved when $\beta$s increased and reached a minimum at $\beta = 0.7$, leading to a 7% reduction in the NRMSE values."*

Also, the authors have missed an opportunity here to deepen their analysis and show the benefits one could retrieve from the use of an hybrid scheme, not only compared to the full static case, $\beta = 0$, but also compared to the full dynamic case, $\beta = 1$. Indeed, the authors do not mention that case while at the same time they show that the hybrid performs better than the EnKF only. What I mean is that if the hybrid was performing better than the static case only but no better than the dynamic case it would be of no interest. So, despite the reference case of the authors being $\beta = 0$, I would highly recommend that they treat the case of the standalone EnKF only for the reason aforementioned and that they complete that paragraph accordingly.

- *We thank the reviewer for this very important suggestion. We propose to add after the sentence in line 271: "Summer 2019, with and without radar QPEs, and winter 2020 have optimal $\beta$ that is equal to 0.5, 0.4, and 0.7, respectively" the following:* *"For all three experiments, the hybrid approach showed its relevance as it overcame both the static ($\beta = 0.0$) and the dynamic configuration ($\beta = 1.0$)."* *We will also further discuss the impact of using $\beta = 1.0$ in Section 7 (see details in the following comment).*

**Sections 7.1 and 7.2:** the authors definitely have to talk more about the case $\beta = 1$. The authors could complete the figures 3 and 4 by adding the curve for $\beta = 1$ and then complete their analysis by emphasizing the fact that the hybrid also improves the results compared to the full dynamic case. I do believe that it would not require too much work from the authors while improving the quality of the paper.

- *This comment was also raised by reviewer #1, and as we agree that taking it into account would benefit the manuscript, we will add more insights on results when $\beta = 1$. The followings provide a list of the modifications we propose:*

  - *Figures 3 and 4 will additionally display the metric (FBI-I, ETS, POD, FAR) values when $\beta = 1$ (grey curve). The new version of these figures and their captions are provided at the end of this document (pages 10-11). We want to draw the reviewer's attention to Figure 3.d. An error occurred while merging different figures. In the current manuscript version, Figure 3.d is the same as Figure 4.b, which is wrong. We will correct Figure 3 accordingly as shown on page 10 of this document. The results are much more consistent with the obtained NRMSE values (Figure 2.b). Indeed, using $\beta = 0.3$ and $\beta = 0.4$ during summer, with the assimilation of radar QPEs, lead to similar verification metrics.*

[revised manuscript text omitted]

**3. Technical corrections**

**Page 3, line 73:** repetition: "the the background field".

- *Thanks you, we will correct this typo.*

**Page 4, line 96**: $P^a{}_{OI}$, is it an error in the notation? Should not it be $P^b{}_{OI}$?

- *We thank the reviewer for this comment. Indeed, we did not introduce $P^a{}_{OI}$ matrix, which corresponds to the covariance matrix of the analysis error when the analysis is solely based on OI approaches. Even if, by design, the time-varying elements of $P^b{}_{OI}$ matrix lead to a time-varying $P^a{}_{OI}$ matrix (see equation 8), we will correct $P^a{}_{OI}$ for $P^b{}_{OI}$ as raised by the reviewer. Introducing a new matrix and its definition will burden the manuscript without adding necessary information.*

**Page 5, line 126:** I would recommend not to write "(1) minus..." but "1 minus". The notation (1) is misleading and can make think about the numerotation of an equation.

- We will take into account this recommendation by writing *"1 minus".*

**Page 6, line 155**: the acronym SYNOP is not defined, does it stand for synoptic?

- *The reviewer is totally right, SYNOP stands for synoptic stations and the acronym was not defined. We intend to modify line 155 p. 6 "For more reliability, the NRMSE is computed only at SYNOP and manual SYNOP stations during the summer and [...]" for "For more reliability, the NRMSE is computed only at surface synoptic observations (hereafter SYNOP) and manual SYNOP stations during the summer and [...]". In addition, as suggested by the other reviewer, we will add an Appendix that lists all the acronyms (including SYNOP).*

**Page 9, eq. (16)**: it seems that there are a few mistakes in the writing of eq. (16), I guess eq. (16) writes:

$$\text{FSS} = 1 - \frac{\frac{1}{N_y} \sum_{i=1}^{N_y} \left( f_a(i) - f_o(i) \right)^2}{\frac{1}{N_y} \left[ \sum_{i=1}^{N_y} f_a^2(i) + \sum_{i=1}^{N_y} f_o^2(i) \right]}$$

- *We thank the reviewer for catching the errors in equation 16. There are indeed one parenthesis that is not right ($f_a(i)$ should have been $f_{a(i)}$) and $n$ in the sum component of the denominator should have been $N_y$. For the other suggestions provided by reviewer #2 (mainly regarding the parenthesis not being in the subscripts), we prefer to follow the same notation as several other articles (see for example equations (2) and (3) in Schwartz et al. 2009). Therefore, we suggest to modify:*

$$FSS = 1 - \frac{\frac{1}{N_y} \sum_{i=1}^{N_y} \left( f_a(i) - f_{o(i)} \right)^2}{\frac{1}{N_y} \left[ \sum_{i=1}^{n} f_{a(i)}^2 + \sum_{i=1}^{n} f_{o(i)}^2 \right]} \tag{16}$$

*for*

$$FSS = 1 - \frac{\frac{1}{N_y} \sum_{i=1}^{N_y} \left( f_{a(i)} - f_{o(i)} \right)^2}{\frac{1}{N_y} \left[ \sum_{i=1}^{N_y} f_{a(i)}^2 + \sum_{i=1}^{N_y} f_{o(i)}^2 \right]} \tag{16}$$

*However, to add more transparency regarding the subscript $i$ definition in equation (16), we will reformulate the following sentence (lines 242-243): "Then, the fractions of grid-cells above the threshold (probabilities) in a pre-selected neighborhood (for example, a square of 30 km) are calculated for CaPA ($f_a$) and ST4 ($f_o$) respectively" for "Then, the fractional values at the ith grid cells (probability of precipitation above a selected threshold) in a preselected neighborhood (e.g., 30 km square) are estimated in CaPA ($f_{a(i)}$) and ST4 ($f_{o(i)}$), respectively."*

**Page 11, line 296**: repetition: "the POD slightly was slightly deteriorated".

- *Thank you, we will rectify that sentence as* *"the POD slightly deteriorated".*

**Page 14, line 425**: repetition: "repetition: "for the observation density observations"".

- *Thank you, we will remove the repetition so that the sentence can read* *"for the observation density".*

**Summer without radars**

[Figure]

**Summer with radars**

[Figure]

**Figure 3**. FBI-1, ETS, POD and FAR across the whole domain for summer experiment without radar QPEs for precipitation analysis with $\beta = 0.0$ (dark blue line), $\beta = 1.0$ (grey line), $\beta = 0.5$ (yellow line in a) and $\beta = 0.3$ (yellow line in b). Same figures but for the summer experiment with the assimilation radar QPEs with $\beta = 1.0$ (grey line), $\beta = 0.4$ (yellow line c) and $\beta = 0.3$ (yellow line d) all three compared to the reference experiment when $\beta = 0.0$ (blue line). Filled markers indicate no significant differences at the 95% confidence level between the reference experiment $\beta = 0.0$ and $\beta = 1.0$, $\beta = 0.5$, $\beta = 0.4$ or $\beta = 0.3$ experiments.

[Figure]

**Figure 4.** Same as Fig. 3 but for the winter experiment and with $\beta = 0.7$ (top panel), $\beta = 0.3$ (bottom panel) and $\beta = 1.0$ (two panels), all three compared to the reference experiment when $\beta = 0.0$.